# A Meta-Analysis of Machine Learning Security Research: Attack-Defense Dynamics, Technical Evolution, and Cross-Concept Patterns

## Abstract

Machine learning (ML) security research has grown rapidly, yet systematic understanding of its technical evolution, attack-defense dynamics, and cross-concept patterns remains limited. This study presents a systematic meta-analysis of 1,591 security papers spanning six security topics and five ML concepts, released between January 1, 2018, and June 30, 2024. Beyond analyzing research trends, we quantify the attack-defense imbalance in research volume across all concept-topic combinations, revealing that defense research significantly lags behind attack research in emerging areas such as LLM jailbreaks (defense ratio = 0.30) and text-to-image membership inference (0.20). Using LLM-assisted annotation of all paper abstracts, we identify 32 distinct technique families and trace their evolution over time, finding that attack techniques such as backdoor injection and adversarial perturbation first appeared, within our corpus, in federated learning and graph neural networks before being adopted in LLM and text-to-image model security research. We further identify 17 technique families shared across multiple ML concepts, with six spanning all five concepts studied. Additionally, we examine factors associated with academic influence, finding that ML concepts, security topics, author count, regions, collaboration patterns, and publication status are all statistically significantly associated with citation density. Our findings highlight critical defense gaps and map the technical landscape of ML security, providing a January 2018 to June 2024 baseline for future longitudinal comparison.

## 1 Introduction

Machine learning (ML) has transformed a wide range of industries, such as healthcare (Wiens & Shenoy, 2018; Binder et al., 2021; Klauschen et al., 2018), finance (Goodell et al., 2021; Dixon et al., 2020), chemistry (Stocker et al., 2020; Keith et al., 2021), and bioinformatics (Senior et al., 2020; ngoc et al., 2020). While the widespread adoption has been transformative, the increasing integration of ML systems into decision-making processes and critical operations has amplified concerns regarding security, fairness, privacy (Xu et al., 2021; Al-Rubaie & Chang, 2019), and the consequences of real-world incidents (The Guardian, 2018). Consequently, understanding and addressing the security of ML systems has become a key focus.

Recent studies have highlighted security threats across various ML paradigms, including federated learning, contrastive learning, large language models (LLMs), text-to-image models, etc. These threats span a broad spectrum of attack vectors, such as model stealing attacks that compromise intellectual property (Tramèr et al., 2016; Shen et al., 2022; Kariyappa et al., 2021; Sanyal et al., 2022), data poisoning attacks that corrupt the training process (Biggio et al., 2012; Chen et al., 2021; Li et al., 2021; Salem et al., 2022), and jailbreak attacks that bypass the security mechanisms of LLMs (Wei et al., 2023; Li et al., 2023; Deng et al., 2023; Shen et al., 2024; Chao et al., 2023). The consequences of these attacks are particularly severe in domains where system integrity is paramount, such as autonomous driving systems (Deng et al., 2021) and healthcare applications (Newaz et al., 2020), where compromised ML models can lead to catastrophic outcomes and significant financial losses.

Such threats have led to substantial research activity in the ML security field, characterized by an ongoing competition between attackers and defenders. Attackers continuously explore novel attack vectors and methods to circumvent existing defenses, while defenders work to design more robust architectures and develop advanced countermeasures (Wu & Wang, 2021; Wang et al., 2020; Huang et al., 2022; Chu et al., 2024; Tramèr et al., 2017). This dynamic interplay has resulted in an exponential increase in academic publications within the ML security field, reflecting the complexity of challenges, the diversity of potential solutions, and the ever-changing nature of the research field.

Despite the surge in research activity, the landscape of ML security research remains largely unknown. Although researchers and practitioners acknowledge the general growth trajectory, and numerous surveys on ML security have been published (Kuntla et al., 2021; Guan et al., 2018), there remains a notable gap in systematically and quantitatively understanding publication patterns, emerging research directions, and the evolution of specific security concerns. This gap limits effective resource allocation, research prioritization, and strategic planning, potentially hindering efforts to address ML security challenges effectively.

This study aims to fill this gap by conducting a systematic meta-study of ML security papers. Specifically, we focus on five major ML concepts: federated learning, contrastive learning, large language models, text-to-image models, and graph neural networks; and six key security topics, including adversarial examples, data poisoning attacks, model stealing attacks, membership inference attacks, jailbreak attacks, and prompt injection attacks.

Our study focuses on three research questions (RQs):

**RQ1:** How has the ML security research landscape evolved over time?

**RQ2:** What technical patterns emerge across ML security research?

**RQ3:** What factors are associated with the academic influence of ML security papers?

**Our Approach.** We collect 1,963 papers from Semantic Scholar (Semantic Scholar, 2024a) and DBLP (DBLP, 2024), covering six security topics across five ML concepts, released between January 1, 2018, and June 30, 2024. After manual review by five domain experts, we retain 1,591 papers. For RQ1, we analyze paper distribution and temporal trends and quantify the attack-defense imbalance across all concept-topic combinations. For RQ2, we use LLM-assisted annotation on all paper abstracts to extract technique families, then trace their evolution over time and identify cross-concept shared technique families. For RQ3, we employ hypothesis testing to evaluate whether six factors (ML concepts, security topics, author count, geographic regions, collaboration patterns, and publication status) are associated with academic influence.[1]

**Main Findings.** The main findings are summarized below, organized by research question.

- The field's growth is uneven: attack research count consistently outpaces defense, and the gap is widest in the fastest-growing areas. For example, LLM jailbreak research has a defense ratio of only 0.30. This suggests that research effort alone does not ensure security readiness (see Section 3).

- We identify 32 distinct families (30 technique families and two analysis/tooling categories). The field is undergoing a structural shift from white-box, training-time settings toward black-box, inference-time settings, where defenses remain scarce. We further find that six technique families span all five ML concepts, suggesting that these technique families (e.g., data poisoning, membership inference) generalize across concepts in our corpus rather than being specific to a single architecture (see Section 4).

- All six factors we examined (ML concepts, security topics, author count, regions, collaboration patterns, and publication status) are statistically significantly associated with citation density. Broader collaboration is associated with greater academic reach (see Section 5).[2]

---

[1]The claim of artifacts and the ethical considerations are addressed in Appendix D and Appendix 10 in the supplementary material, respectively.

[2]We do not believe citation metrics should drive research priorities. However, understanding the factors correlated with academic influence can help researchers make informed decisions about collaboration and dissemination strategies.

Table 1: Five ML concepts studied in this work.

| Concept | Abbr. | Description |
|---|---|---|
| Federated Learning | FL | A distributed ML training strategy designed to enhance privacy protection (McMahan et al., 2016). |
| Contrastive Learning | CL | An unsupervised learning method that learns data representations by comparing similarities and differences between samples (Chen et al., 2020; Coates et al., 2011). |
| Large Language Model | LLM | A pre-trained model capable of understanding various types of data and generating human-like responses, including both unimodal and multimodal models (OpenAI, 2023; Yin et al., 2023). |
| Text-to-Image Model | T2IM | A generative model that produces images from natural language descriptions. |
| Graph Neural Network | GNN | A neural network designed to process graph-structured data (Scarselli et al., 2009). |

Table 2: Six security topics studied in this work. Our study covers both attacks and their corresponding defenses.

| Topic | Abbr. | Description |
|---|---|---|
| Adversarial Example | AE | Inputs subtly perturbed to cause incorrect model predictions (Goodfellow et al., 2014). |
| Data Poisoning | DP | Manipulating training data to alter model behavior; backdoor attacks (Chu et al., 2024) are a representative form (Biggio et al., 2012). |
| Model Stealing | MS | Extracting model information (weights, architecture) through query interactions (Tramèr et al., 2016). |
| Membership Inference | MIA | Inferring whether a data point was used in model training (Shokri et al., 2017; Choo et al., 2021). |
| Jailbreak | JB | Manipulating model input to bypass safeguards and elicit prohibited content (Wei et al., 2023; Li et al., 2023). |
| Prompt Injection | PI | Manipulating input to cause the model to deviate from its intended purpose (Greshake et al., 2023; Perez & Ribeiro, 2022). |

## 2 Research Data

### 2.1 Research Scope

Machine learning is a rapidly evolving field, with new concepts, spanning algorithms, frameworks, and models, being introduced continuously, leading to consistent improvements in model performance. However, high-performance models are accompanied by growing security and privacy risks (Tramèr et al., 2016; Shen et al., 2024; Zhang et al., 2023). Consequently, researchers have devoted significant efforts to studying both attacks on and defenses for these concepts. While we acknowledge that it is impossible to cover all ML concepts and security topics in such a flourishing research domain with limited human resources, our study focuses on those proposed between January 1, 2018, and June 30, 2024, as they are typically prioritized by the recent research community and are more likely to yield statistically significant conclusions. Specifically, our study focuses on papers about six security topics against five ML concepts.

We focus on five ML concepts that are widely used and increasingly exposed to real-world security threats (Table 1), and six security topics covering the major types of attacks and corresponding defense mechanisms (Chowdhury et al., 2024; Rosenberg et al., 2021; Paracha et al., 2024) (Table 2). We do not claim these lists are exhaustive or mutually exclusive; the rationale for this selection and operational definitions for each topic, including how overlapping cases are assigned, are provided in Appendix A.

Note, in this study, we focus more on how the ML concepts are targeted by the security topics, rather than how security topics are implemented using ML concepts. For instance, regarding a paper that uses Contrastive Learning (CL) to generate adversarial examples (AE) against Text-to-Image Models (T2IMs), the ML concept of the paper is "T2IMs" rather than "CL."

### 2.2 Data Sources

We rely on two academic search engines as the data sources.

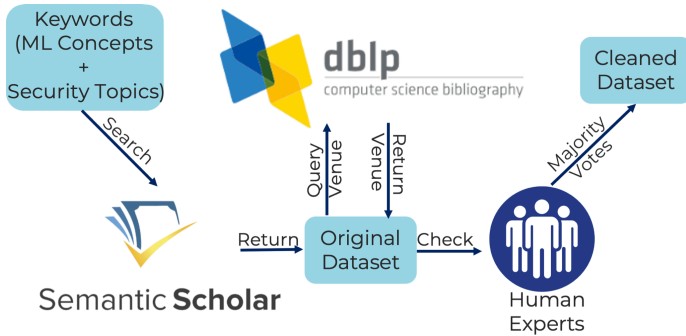

Figure 1: Dataset collection workflow.

**Semantic Scholar (Semantic Scholar, 2024a).** This service has a collection of over 200 million publications across a wide range of scientific fields, offering extensive coverage of papers from various publishers and preprint databases. Moreover, it supports match query functionality (Semantic Scholar, 2024b), enabling us to obtain related papers by providing corresponding keywords.

**DBLP (DBLP, 2024).** Certain attributes, such as the publication venue, are occasionally missing from the papers retrieved via the Semantic Scholar API. To address this issue, we integrate the DBLP API to retrieve the missing attributes for these papers. Additional details regarding these two data sources are provided in Appendix B.

### 2.3 Data Collection

In this section, we first introduce our data collection workflow and then elaborate on each step in the workflow.

**Overview.** The overall data collection workflow is presented in Figure 1. The process begins with the bulk matching query of paper titles and abstracts, using various keywords through the Semantic Scholar API to identify related publications. We then use the DBLP API to fill in the missing venue attributes of the collected papers. Upon completing these steps, we obtain an original dataset. We then conduct a human review process to filter out papers that do not meet the requirements and augment the data with attributes such as institutes. The paper collection and screening process follows the PRISMA (PRISMA, 2020) guideline, as illustrated in Figure 23 (Appendix G).

**Publications Collection via Semantic Scholar API.** To systematically collect relevant papers for our study, we first developed keyword sets for the five ML concepts and the six safe topics, respectively. These keyword sets are then used to generate 30 sets of keyword pairs by combining the keyword sets for the five ML concepts with those for the six security topics in a pairwise manner. The details of the keyword set selection process, the entire keyword sets, and the pairing strategy are provided in Appendix A. These keyword pair sets enable us to leverage the bulk-matching query function of the Semantic Scholar API (Semantic Scholar, 2024b) to identify relevant papers. The bulk matching query function of the Semantic Scholar API can retrieve relevant papers within a specified time range and support paging to handle large query results. We utilized the Semantic Scholar API to query these 30 sets of keyword pairs sequentially, setting the search period from January 1, 2018, to June 30, 2024, to obtain 30 subdatasets. In each subdataset, we labeled every entry with "MLConcept" and "SecurityTopic" based on the keyword pair used in the query. For instance, in a subdataset obtained using the keyword pair ("LLM," "MS"), entries were labeled with "MLConcept" as "LLM" and "SecurityTopic" as "MS." Together, these 30 subdatasets constituted our original dataset.

**Completion of Missing Publication Venues via DBLP API.** Among the data obtained from Semantic Scholar, we found 84 papers missing their publication venues. To address this issue, we use DBLP to fill in the missing venues. Specifically, we queried each paper's title to retrieve its publication venue by DBLP Search API (DBLP, 2024) and restricted the API to return only the single most relevant result. We then verified the accuracy of the retrieved data by comparing the authors in the DBLP result with the authors listed for

Table 3: The distribution of the final cleaned dataset (1,591 unique papers; 1,682 concept×topic entries).

| Security Topics \ ML Concepts | Federated Learning | Contrastive Learning | Large Language Model | Text-to-Image Model | Graph Neural Network | Total |
|---|---|---|---|---|---|---|
| Adversarial Example (AE) | 48 | 30 | 55 | 38 | 33 | 204 |
| Data Poisoning Attack (DP) | 706 | 35 | 82 | 29 | 64 | **916** |
| Model Stealing Attack (MS) | 76 | 6 | 11 | 3 | 17 | 113 |
| Membership Inference Attack (MIA) | 111 | 5 | 23 | 20 | 45 | 204 |
| Jailbreak Attack (JB) | - | - | 200 | 5 | - | 205 |
| Prompt Injection Attack (PI) | - | - | 40 | - | - | 40 |
| **Total** | **941** | 76 | 411 | 95 | 159 | 1,682[1] |

[1] Some papers cover multiple topics, thus they are recorded multiple times in the dataset, causing the total number to be higher than 1,591.

the corresponding paper in the original dataset. If the authors matched, we deemed the result accurate and updated the missing venue information accordingly. After processing all 84 papers, we successfully retrieved publication venues for 13 papers. The remaining 71 papers either lacked venue information in DBLP or were not found in the database. To avoid potential bias introduced by excessive manual intervention, we omit them in the final dataset.

**Human Annotation.** The original dataset, which comes from Semantic Scholar and is supplemented by DBLP, may contain some papers of low relevance. For instance, a paper on industry security might mention backdoor attacks as a contextual background, even though its primary focus is not on backdoor attacks. Specifically, we engage five human experts, each either pursuing or holding a Ph.D. degree in computer science and having at least one year of professional experience in ML security, to perform human annotation tasks. For each paper, three experts independently review the title and abstract to verify its pre-assigned "MLConcept" and "SecurityTopic" labels, and then annotate whether the paper should be kept ("IfKeep") and whether it belongs to the attack or defense category ("AttackOrDefense"). Benchmarks, surveys, and systemization of knowledge papers are excluded from our study. Each paper is annotated three times by different experts, and the final label is determined by majority vote. The annotation demonstrates a substantial inter-agreement among the labelers, with the corresponding Fleiss' Kappa (Falotico & Quatto, 2015) statistics reported in Table 15 in Appendix G. Since the "MLConcept" and "SecurityTopic" labels are deterministically derived from the query keyword pairs rather than independently judged by the annotators, they are not subject to inter-annotator disagreement; we therefore report Fleiss' Kappa only for the two attributes that are genuinely produced by human annotation, namely "IfKeep," and "AttackOrDefense." The threat-model dimensions (e.g., access level and attack phase) are instead produced by the LLM-assisted annotation pipeline, whose reliability is evaluated in Appendix E separately. In addition, human experts manually review each paper to identify affiliated institutes and regions. Our entire human annotation process costs over 300 person-hours.

**Data Statistics.** Leveraging Semantic Scholar and DBLP, we collected 1,963 security papers released from January 1, 2018, to June 30, 2024, and their related metadata as the original dataset, including the titles, abstracts, venues, available dates, authors, and citation counts, covering six security topics targeting five ML concepts. After human annotation, we finalized the cleaned dataset comprising 1,591 unique papers. The detailed paper distribution of the final cleaned dataset is shown in Table 3. The overview of the attribute of the cleaned dataset is provided in Table 13 in Appendix G.

**Counting Conventions.** To avoid ambiguity across tables, we use two counting conventions and state which one applies in each caption. The dataset has 1,591 unique papers but 1,682 concept-topic entries, because a paper that addresses more than one concept-topic combination contributes one entry per combination (the 91 extra entries arise from such papers). Entry-based tables (for example, the concept-topic distribution and the technique-family entry counts) therefore sum to 1,682, whereas unique-paper tables sum to 1,591. Regional counts use a third convention: a paper is attributed to every region of its authors, so the per-region totals exceed 1,591 (534 papers have authors from more than one region, and the per-region unique-paper counts sum to 2,310).

# 3 Research Landscape and Attack-Defense Dynamics (RQ1)

In this section, we investigate the distribution of security papers across different concepts and how the focus of security topics targeting different ML concepts has evolved over the past six and a half years. We first analyze the distribution from both static and temporal perspectives, then examine the attack-defense balance across all concept-topic combinations to identify areas where defense research is sparsest relative to attack research.

## 3.1 Paper Distribution and Temporal Trend Across ML Concepts

**Overall Results.** As shown in Table 3, among all ML concepts we study, federated learning has the largest number of papers, with 941 papers, followed by LLMs with 411 papers. The least studied ML concept is contrastive learning, with only 76 papers, less than 8.3% of federated learning. These counts describe our corpus rather than the field as a whole. We count papers in specific concept-topic combinations where one of our five ML concepts is the attack target, so a large body of security research on other model types, for example multilayer perceptrons or generic classifiers, is outside our scope and not reflected here. The distribution and temporal figures in this section are therefore descriptive of our corpus under our scope and definitions, and are not claims about the size of the field or the real-world security of any system.

**Paper Distribution of Five ML Concepts.** We find that for different ML concepts, the most popular security topic against them varies, as shown in Table 3. Specifically, data poisoning attacks are the most popular security topic against federated learning, contrastive learning, and graph neural networks, with 706, 35, and 64, respectively. The security papers for data poisoning attacks against the above three ML concepts account for 75.0%, 46.1%, and 40.3% of the total number of security papers against them separately. Regarding LLMs, jailbreak attacks are particularly prominent, with the highest number of security papers (200), which makes up 48.7% of the total number of security papers against LLMs. For text-to-image models, adversarial examples remain the most studied, with 38 papers representing 40.0% of the total.

**Temporal Trends of Five ML Concepts.** In Figure 2, we illustrate the temporal trends in the number of new security papers against ML concepts per quarter from January 1, 2018, to June 30, 2024. The number of new security papers per quarter on all five ML concepts has been steadily increasing from January 1, 2018, to June 30, 2024.

Specifically, the number of new security papers related to federated learning per quarter has continuously grown since 2019, with a rapid surge beginning in 2021. From 2018 until the end of 2023, federated learning consistently held the top position in terms of the number of new security papers per quarter. Research on LLMs began gaining significant attention in 2022 and witnessed a sharp surge in the same year. By 2024, the number of new papers in this topic surpassed that of federated learning, making it the most researched ML concept. Meanwhile, the number of new security papers per quarter on text-to-image models, graph neural networks, and contrastive learning has shown steady yet relatively modest growth, with a more gradual upward trend compared to the other two concepts.

**Temporal Trends of Six Security Topics Against Each ML Concept.** Next, we perform a more detailed analysis of how the number of new security papers of six security topics per quarter has evolved over time on each ML concept, with the results shown in Figure 3 and Figure 24 in Appendix G.

*Federated Learning.* Figure 2b shows the trend in the number of new security papers of federated learning from 2018 to 2024, categorized by four types of attacks: adversarial examples, backdoor attacks, model stealing, and membership inference attacks. Data poisoning attacks dominated the number of new security papers per quarter, starting to rise significantly around 2021 and accelerating sharply after 2022. By 2024, the number of new papers per quarter in this topic has exceeded 83, far exceeding the other topics. The other three security topics, i.e., adversarial examples, model stealing, and MIA, had relatively stable and small numbers of new papers, fluctuating between 0 and 12 papers over the years. While MIA had occasional spikes, it was still close to the level of adversarial examples and model stealing. None of the three methods has seen the same explosion as poisoning and backdoor attacks. These data suggested that backdoor attacks were clearly gaining attention in the context of federated learning, especially in the last few years.

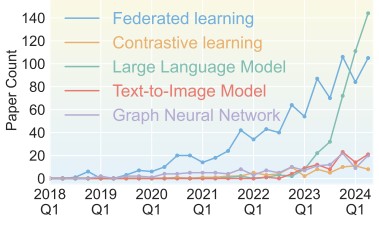
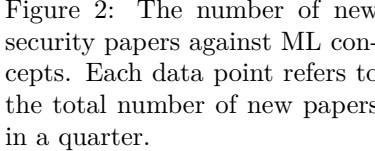

Figure 2: The number of new security papers against ML concepts. Each data point refers to the total number of new papers in a quarter.

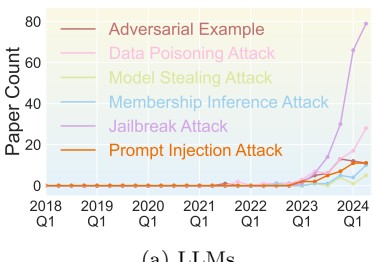

(a) LLMs

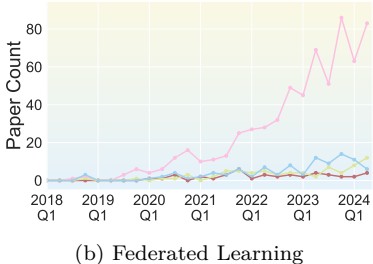

(b) Federated Learning

Figure 3: The number of new papers on selected ML concepts (LLMs and FL) per quarter. Each data point refers to the total number of new papers in a quarter.

*LLMs.* Figure 2a illustrates the trends in the number of new security papers focusing on LLMs per quarter from 2018 to the first half of 2024. Jailbreak attacks exhibit the most significant growth, particularly starting in 2023. In the second quarter of 2024, the number of new jailbreak papers against LLMs per quarter reached 79, far exceeding papers on other security topics. Data poisoning attacks also experienced notable growth, becoming the second most discussed topic by 2024, with 28 papers. The other four attack methods, i.e., adversarial examples, model stealing, MIA, and prompt injection, showed an increase in the number of new security papers per quarter starting in 2023, but their growth was relatively moderate compared to the sharp rise seen in jailbreak and backdoor attacks.

*GNN, CL, and T2IMs.* The trends of GNN, CL, and T2IMs are illustrated in Figure 24 in Appendix G. Overall, we observed that GNN demonstrated a similar trend to that of federated learning. For contrastive learning and text-to-image models, the number of new security papers per quarter on various security topics began to increase in 2020 and 2022, respectively. However, due to the relatively small number of papers, no clear trends can be observed.

In summary, federated learning consistently dominated security research while LLMs surged a lot. This is potentially driven by data privacy regulations like GDPR (European Union, 2016), which came into effect in 2018. However, driven by ChatGPT's public release and subsequent widespread adoption, LLM security papers have surged in 2023, overtaking federated learning in Q1 2024 (see Figure 2). This surge also indirectly reflects the popularity of LLMs themselves.

## 3.2 Paper Distribution and Temporal Trend Across Security Topics

**Overall Results.** As shown in Table 3, among all the security topics, papers related to data poisoning attacks are the most numerous, totaling 916. Adversarial examples, membership inference attacks, and jailbreak attacks each have a similar number of papers, around 200. Model-stealing attack papers are relatively fewer, with only 113, despite spanning all five ML concepts. Additionally, prompt injection attack papers have the smallest amount, with only 40, as this topic is exclusively associated with LLMs.

**Paper Distribution of Six Security Topics.** From Table 3, it could be easily found that for data poisoning attacks, model stealing, and membership inference attacks, federated learning emerged as the most popular topic. The number of papers for data poisoning attacks, model stealing, and membership inference attacks against federated learning was 706, 76, and 111, accounting for 77.1%, 67.3%, and 54.4% of the total, respectively. In addition, federated learning had a dominant position in these three security topics, and its number of papers far exceeded that of other ML concepts. In contrast, the most popular ML concept for the remaining three attack methods (adversarial examples, jailbreak, and prompt injection) was LLMs, with 55, 200, and 40.

**Temporal Trend of Six Security Topics.** Figure 4 shows the overall trends in the number of new security papers per quarter related to six security topics. The number of new papers per quarter on data poisoning attacks has experienced the most significant growth over time, with a noticeable increase starting around

2020. By the second quarter of 2024, it reached the highest number, exceeding 132. From the second quarter of 2019 to the second quarter of 2024, it has been the most popular security topic, with a much higher number of papers than other security topics. Research on jailbreak attacks began to surge rapidly in 2023, and by 2024, this security topic had become the second most studied attack method, with the number of new papers per quarter surpassing 82 from the first quarter of 2024. The number of new papers per quarter on adversarial examples, model stealing, and MIA has shown steady but relatively slower growth over the years. As for prompt injection, this type showed a slight increase in the number of new security papers per quarter starting in 2023, but compared to poisoning and jailbreak attacks, it remains one of the less explored topics.

**Temporal Trends of Each Security Topic Against Five ML Concepts.** We also performed a more fine-grained analysis of how the number of new security papers of each ML concept per quarter has evolved over time on each security topic. The related results are shown in Figure 5 and Figure 25. These figures reveal which ML concepts have become increasingly popular under each security topic over time.

*AE.* First, Figure 4a illustrates the trend of the number of new security papers focusing on adversarial examples against five ML concepts per quarter from 2018 to 2024. The number of new security papers per quarter related to adversarial examples against LLMs began to gain traction in 2022 and show rapid growth thereafter, becoming the most researched topic from the second quarter of 2023. Research on adversarial examples against text-to-image models began to increase significantly around 2023 and has been on a steady upward trend since then. The number of new security papers per quarter on adversarial examples in federated learning started appearing sporadically in 2020 and reached a significant peak of activity around 2021 to 2023. As for the remaining two ML concepts to adversarial examples, both showed limited but sustained research activity with occasional spikes, but both remain relatively less explored compared to LLMs and text-to-image models.

*DP.* In the context of the number of new security papers per quarter for data poisoning attacks from 2018 to 2024, federated learning has experienced rapid growth, particularly since 2021, consistently surpassing other ML concepts and remaining the most prominent topic in the field. LLMs have shown significant growth since 2023, becoming the second most popular topic by the latter half of the year, following federated learning. The other three ML concepts have also seen a gradual increase in attention, but their number of new security papers per quarter remains considerably lower than the top two.

*MS and MIA.* Similarly, for model stealing and membership inference attacks shown inin Figure 25 in Appendix G, although the number of new papers per quarter was far from that of data poisoning attacks, the overall trend was consistent with data poisoning attacks.

*JB and PI.* Finally, regarding jailbreak and prompt injection, we found almost only papers against LLMs for both. This also implied the specificity of these two security topics. Despite their differences in number, these two topics exhibited similar trends. Starting in 2023, the number of new security papers per quarter began to rise significantly, reaching a peak in 2024. This indicates that, with the widespread adoption of LLMs, the potential risks posed by jailbreak and prompt injection attacks have increasingly drawn the attention of the academic community.

In summary, data poisoning attacks remain the most researched security topic. Given ML models' reliance on large, uncurated datasets, this highlights the diverse range of attack vectors associated with these threats and their significant real-world implications (e.g., Google incident (Google, 2024)). Security research on federated learning predominantly focuses on data poisoning attacks, whereas studies on LLMs concentrate on jailbreak attacks.

### 3.3 Paper Distribution of Publication Statuses

Finally, we analyze which ML concepts of security papers are more likely to be published, and which security topics have higher publication rates for security papers. The publication rate in our study is defined as the percentage of published papers on a topic relative to the total number of papers on the same topic between January 1, 2018, and June 30, 2024. And we defined preprints as unpublished papers and papers published in scientific journals or conferences as published papers. First, with respect to which ML concepts attack

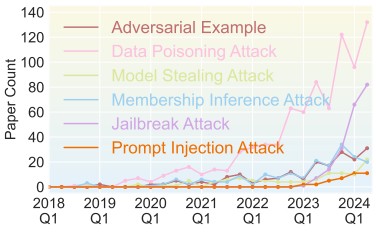

Figure 4: The number of new security papers on security topics. Each data point refers to the total number of new papers in a quarter.

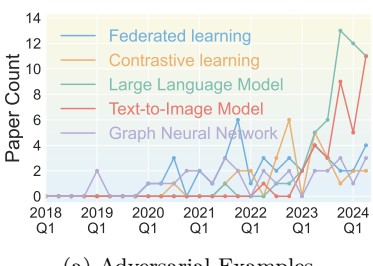

(a) Adversarial Examples

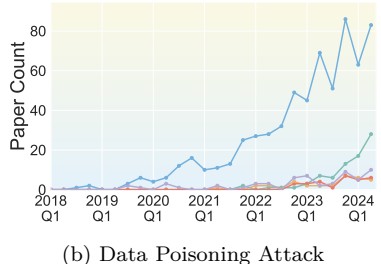

(b) Data Poisoning Attack

Figure 5: The number of new papers on selected security topics (AE and DP) against each ML concept per quarter. Each data point refers to the total number of new papers in a quarter.

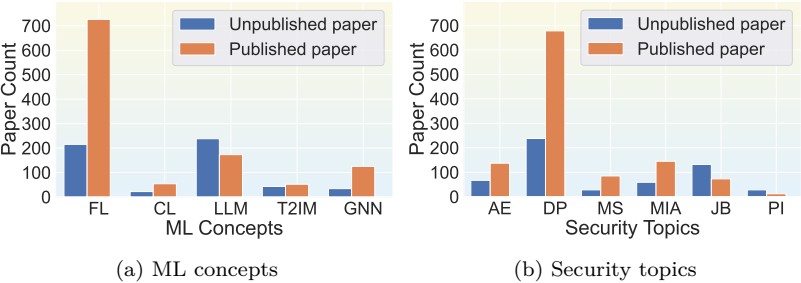

(a) ML concepts

(b) Security topics

Figure 6: The number of published versus unpublished papers.

and defense papers are more likely to be published, Figure 6a shows that contrastive learning has the highest publication rate (0.786) among the five ML concepts, followed by GNNs. Federated learning also has a high publication rate of 0.772, but slightly lower than contrastive learning. Large language models are the only ML concept where the number of unpublished papers exceeds the number of published papers.

Figure 6b illustrates the comparison of the number of unpublished papers to the number of published papers against the six attack methods. Among these, model stealing has the highest rate of published papers (0.752). Following this, data poisoning ranks second, with a slightly lower publication rate of 0.740. In contrast, jailbreak and prompt injection are the only two attack methods where the number of unpublished papers exceeds the number of published papers, with published rates of 0.356 and 0.300, respectively.

LLM security papers, despite their high volume, show the lowest publication rate, particularly in jailbreak and prompt injection attacks. One possible explanation is that the current work is still in its early stages and has not been sufficiently refined, or that the vulnerabilities studied have been quickly fixed, leading to the community's lack of widespread acceptance that it is sufficient for publication. However, this interpretation remains speculative and warrants further investigation.

### 3.4 Imbalance in Attack/Defense Research Effort

Beyond counting the total number of papers, we analyze the balance between attack and defense research across all concept-topic combinations. For each combination, we compute the *defense ratio*, defined as the number of defense papers divided by the total number of papers in that combination. Papers labeled as "Both" (containing both attack and defense contributions) are counted toward both categories. We verified that the per-cell defense ratios are robust to this convention: recomputing under three alternative rules (excluding "Both" papers, counting them only as defense, or counting them only as attack) changes each ratio by at most 0.05 and preserves the set of most under-defended combinations (Appendix C). We emphasize that the defense ratio and the defense lag measure the relative allocation of research effort and attention across combinations. They are proxies for where the community has invested, and they do not

measure the empirical robustness of deployed systems or the relative strength of individual attacks and defenses.

**Overall Results.** Figure 7 presents the defense ratio for all 30 concept-topic combinations. Several findings stand out. First, federated learning exhibits relatively balanced attack-defense research, with defense ratios above 0.50 for most security topics (e.g., FL×DP = 0.74, FL×MS = 0.87). Second, LLM security research is significantly skewed toward attacks: LLM×JB has a defense ratio of only 0.30, LLM×DP = 0.31, and LLM×MIA = 0.26. Third, similar imbalances appear for text-to-image models (T2IM×MIA = 0.20, T2IM×DP = 0.28) and graph neural networks (GNN×DP = 0.30, GNN×AE = 0.36).

In summary, a systematic attack-defense imbalance exists in ML security research. Emerging areas (LLMs, T2IMs) have defense ratios below 0.40, while mature areas (FL) have ratios above 0.55.

**Defense Lags.** We also quantify the *defense lag*, defined as the number of quarters between the first attack paper and the first defense paper for each combination. Table 21 (Appendix G) summarizes the defense lag for all non-empty concept-topic combinations. Notable positive defense lags, where defense trails attack, include CL×DP (7 quarters), GNN×MS (8 quarters), LLM×DP (5 quarters), and T2IM×JB (4 quarters), indicating that defense research often takes substantial time to respond to newly emerging attack vectors. Part of this lag reflects a structural asymmetry in the burden of proof. Validating an attack usually requires demonstrating success against a specific model or scenario, whereas a credible defense must withstand a range of attacks and generalize across architectures and threat models. This higher bar raises the barrier to entry for defense work, so some lag between the first attack and the first defense is expected and does not by itself indicate community neglect.

Several combinations, however, exhibit *negative* defense lags, meaning that papers categorized as defenses appeared before the corresponding attack papers. Rather than suggesting that defense systematically outpaced attack, this pattern indicates that many protective mechanisms were initially developed under broader or adjacent security objectives and were only later associated with a more clearly defined attack category. For example, in FL×MS (lag = −13), differential privacy and secure aggregation were introduced as general privacy protections for federated learning as early as 2018, well before model stealing attacks against FL were explicitly studied in 2022. A similar pattern appears in LLM×MS (lag = −2), where watermarking methods for copyright protection preceded explicit studies of model extraction attacks. More broadly, this finding suggests that effective defenses often emerge not from reacting to a single attack formulation, but from developing transferable protection mechanisms that remain useful across evolving and partially overlapping threat boundaries. This observation is consistent with the value of generalizable defenses. As noted above, the negative lags partly reflect labeling and topic-boundary effects, so we read this as suggestive rather than as evidence that defense systematically outpaces attack.

**Potential Research Opportunities.** Figure 8 provides an integrated view by plotting each concept-topic combination along four dimensions: paper volume, defense ratio, recent growth rate, and publication rate. This research opportunity map reveals three distinct regions: (1) mature combinations with high volume and balanced attack-defense research (e.g., FL×DP); (2) rapidly growing but defense-deficient combinations (e.g., LLM×JB); and (3) sparsely studied combinations that represent potential research gaps (e.g., CL×MS with a defense ratio = 0, T2IM×MS with only 3 papers).

These patterns suggest that future work should prioritize not only publication volume, but also the structural imbalance between attack development and defense research across combinations. In particular, the second region appears especially promising for high-impact research, because rapid growth coupled with low defense coverage indicates that threat awareness is advancing faster than defensive understanding. For these combinations, important next steps include developing more realistic evaluation protocols and more generalizable defense mechanisms rather than isolated point solutions. By contrast, the first region is less likely to benefit from further incremental gains alone and may instead require consolidation-oriented efforts such as standardized benchmarks, reproducibility studies, and deployment-aware evaluation. Meanwhile, the third region offers opportunities for problem-defining research, where the establishment of clear attack surfaces, realistic application scenarios, and initial baselines may be as valuable as proposing new methods.

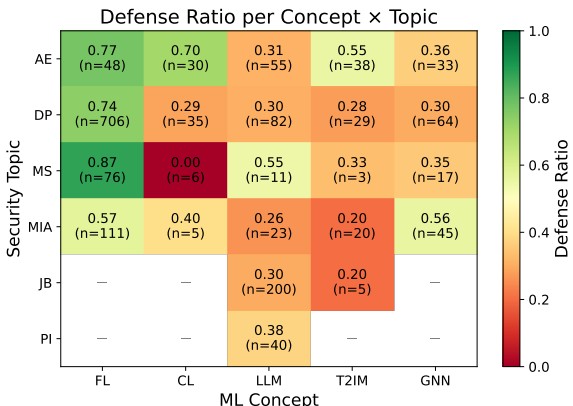
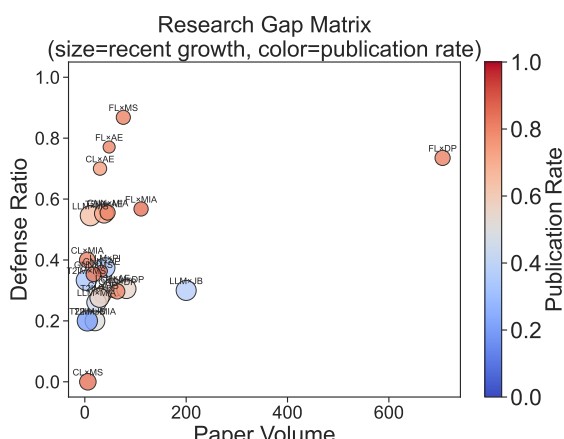

Figure 7: Defense ratio for each concept×topic combination. Green indicates balanced research; red indicates attack-dominated areas.

Figure 8: Research opportunity map. Each point represents a concept×topic combination. *x*-axis: paper volume; *y*-axis: defense ratio; point size: recent growth (2023–2024); color: publication rate.

### 3.5 Takeaways

**First, defense research should be prioritized in areas where the attack-defense gap is widest.** The community's current research effort is unevenly distributed. Some combinations have accumulated hundreds of attack papers with minimal defensive counterparts. Directing new research toward these under-defended areas would address the widest gaps in defensive research coverage. **Second, broadly scoped protection mechanisms can sometimes provide coverage before a specific attack is formalized.** Our defense lag analysis shows that some protection mechanisms appeared before the corresponding attacks were formally defined. We caution that these negative lags also reflect labeling and topic-boundary effects rather than proactive design alone, so we do not claim that proactive defense is more effective than reactive response. Even so, given the structural asymmetry that makes defenses harder to validate than attacks, investing in generalizable defense frameworks that remain effective across evolving threat boundaries is a reasonable strategy. **Third, the ongoing shift from training-time to inference-time threat models demands new defensive thinking.** The techniques, assumptions, and evaluation norms established in the FL-dominated era are rooted in white-box, training-time settings. As the field moves toward black-box, inference-time scenarios driven by LLM deployment, researchers should not simply adapt existing training-time defenses but develop fundamentally new approaches suited to inference-time constraints.

## 4 Technical Evolution and Cross-Concept Patterns (RQ2)

To move beyond paper-level metadata analysis, we investigate the technical details employed in the 1,591 papers, such as the threat model and the technique families. Since a paper may be relevant to multiple concept×topic combinations, the dataset contains 1,682 entries. Unless otherwise noted, the counts reported in this section refer to unique papers (1,591 total). Within specific concept×topic combinations (e.g., FL×DP), counts refer to entries in that combination.

### 4.1 Threat Model Landscape and Temporal Shift

**Landscape.** The threat model assumed by each paper, such as the *access level* (white-box, black-box, or gray-box) and the *attack phase* (training-time or inference-time), reveals fundamental structural patterns in ML security research. Of the 1,591 papers, 893 (56.1%) assume a white-box setting, 439 (27.6%) black-box, and 126 (7.9%) gray-box, with 149 (9.4%) classified as not applicable (e.g., defense-only papers where access level is not defined). Regarding the attack phase, 1,098 (69.0%) target the training phase and 517 (32.5%) target inference time. Papers addressing both settings (16 for access, 31 for phase) are counted

Table 4: Attack-defense statistics by threat model category. Papers addressing "both" settings are counted in both corresponding quadrants. DefR = defense papers / total papers.

| Quadrant | Total | Attack | Defense | DefR |
|---|---|---|---|---|
| White-box × Training | 777 | 294 | 473 | 0.61 |
| Gray-box × Training | 117 | 36 | 78 | 0.67 |
| White-box × Inference | 136 | 72 | 55 | 0.40 |
| Black-box × Training | 74 | 45 | 28 | 0.38 |
| Black-box × Inference | 372 | 279 | 80 | **0.22** |

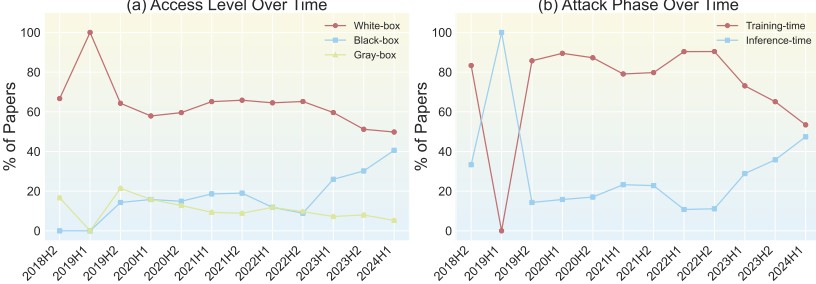

Figure 9: Temporal shift in threat model distribution. (a) Access level: white-box research is declining while black-box research is growing rapidly; gray-box remains a niche FL-specific setting. (b) Attack phase: training-time research is giving way to inference-time research. Papers addressing "both" settings are counted in both categories.

in both corresponding categories; 7 papers have no applicable phase. Combining these two dimensions yields a five-cell view (Table 4), revealing two dominant research clusters with very different attack-defense characteristics:

- **Cluster A (training-time, white/gray-box)** encompasses 894 papers (777 white-box + 117 gray-box) with a defense ratio of 0.62. This cluster is dominated by FL security research and features a mature defense ecosystem with five major defense families: anomaly detection, robust aggregation, differential privacy, adversarial training, and federated secure aggregation. Gray-box papers (126 total) are almost exclusively FL-related (94%), reflecting FL's unique threat model where a participating client has white-box access to its local model but limited information about the global model.

- **Cluster B (inference-time, black-box)** contains 372 papers with a defense ratio of only 0.22. This cluster is dominated by LLM security research and features 13 distinct attack families but only two major defense families: safety filters and input preprocessing. The attack-to-defense imbalance in this cluster is the most severe in the dataset.

**Temporal Shift.** The balance between these two clusters has shifted dramatically over the study period. As shown in Figure 9(a), white-box research declined from 65% of all papers in 2022 to 50% in 2024, while black-box research grew from 10% to 41%. Gray-box research, which is almost exclusively associated with FL (94% of gray-box papers), declined from 18% in 2018–2019 to 5% in 2024 as LLM and T2IM research grew and diluted its share. Similarly, Figure 9(b) shows that training-time research declined from 90% in 2022 to 53% in 2024, while inference-time research grew from 11% to 47%. This shift coincides with the rise of LLM security research: LLMs are predominantly accessed as black-box APIs, and security concerns such as jailbreaks and prompt injection arise at inference time.

The transition has important implications: the defense techniques developed for Cluster A (robust aggregation, anomaly detection of malicious training updates) are largely inapplicable to Cluster B, where defenders can only intervene at the input/output level. As shown in Figure 10(a), the defense ratio in white-box settings has remained stable at 0.55–0.70, while the defense ratio in black-box settings has stagnated at

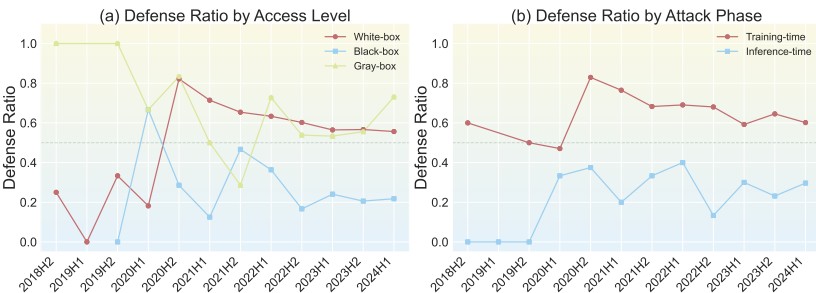

Figure 10: Defense ratio over time by threat model dimension. The dashed gray line marks 0.50 (parity between attack and defense). (a) By access level: white-box and gray-box maintain relatively balanced defense ratios, while black-box remains low. (b) By the attack phase, training-time defense is stable, inference-time defense shows slow recovery but remains far from parity.

0.20–0.25 despite rapid growth in research volume (from 22 papers in 2022 to 195 in 2024). Gray-box settings maintain a relatively high defense ratio (0.60–0.70), reflecting the maturity of FL's defense ecosystem. Figure 10(b) shows a similar pattern by attack phase: training-time defense ratio remains stable around 0.60, while inference-time defense ratio shows a modest recovery from 0.23 in 2022 to 0.29 in 2024, but remains far below parity.

## 4.2 Technique Family Distribution and Evolution

We introduce the notion of a *technique family*, which groups papers by the specific attack or defense methodology they employ. For example, within the broad category of "data poisoning attacks," we distinguish between *backdoor injection* (planting a hidden trigger during training), *general data poisoning* (corrupting training data without a trigger), and *model inversion* (reconstructing training data from model outputs). Similarly, on the defense side, we distinguish between *robust aggregation* (Byzantine-tolerant aggregation in federated learning), *anomaly detection* (detecting poisoned data or malicious clients), *differential privacy* (adding noise for privacy guarantees), and others. This finer-grained classification enables us to track how specific technical approaches have evolved over time and which approaches are shared across ML concepts.

To assign technique families at scale, we use LLM-assisted annotation (Claude Haiku 4.5) on all paper abstracts. For each paper, we extract the primary technique family from a predefined taxonomy of 30 categories, as well as coarse-grained threat model dimensions (white-box vs. black-box, training-time vs. inference-time). The annotation quality was verified through human review, achieving an accuracy above 93%. Details of the annotation taxonomy, prompt, and quality report are provided in Appendix E.

### 4.2.1 Overall Technique Family Distribution

Across the entire dataset, we identify 32 distinct families (30 technique families and two other families), with the top 10 technique families summarized in Table 5. The distribution reveals that *defense-oriented* technique families collectively outnumber attack-oriented ones. The top three are anomaly detection (234 papers), robust aggregation (204), and backdoor injection (197), reflecting the dominance of data poisoning as a security topic (see Section 3.2) and the maturity of FL's defense ecosystem.

### 4.2.2 Technique Family Evolution

For each concept-topic combination with at least 20 papers, we trace the temporal evolution of technique families over time, yielding 17 evolution charts in total. To better highlight broader ecosystem-level patterns, we organize the representative cases into two groups: federated learning (FL), which generally exhibits more mature and balanced attack-defense development, and large language models (LLMs), where rapid growth is often accompanied by stronger concentration and weaker defensive diversification. The remaining charts are provided in Appendix G.

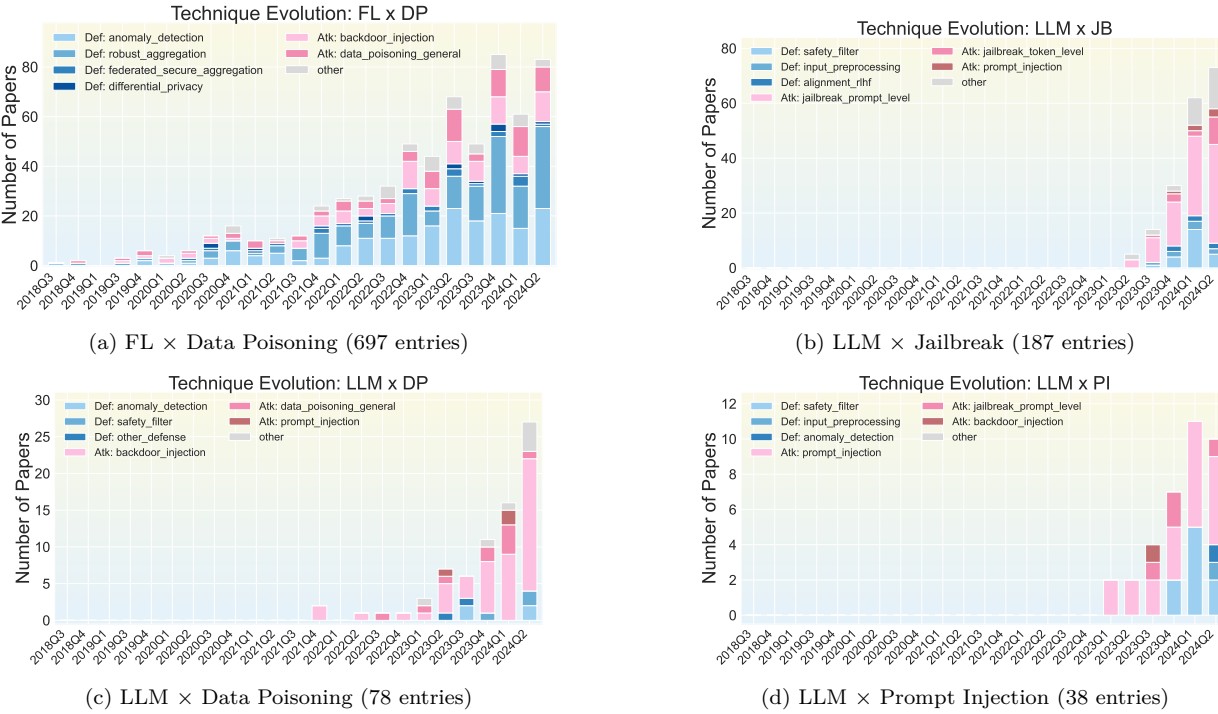

(a) FL × Data Poisoning (697 entries)  (b) LLM × Jailbreak (187 entries)

(c) LLM × Data Poisoning (78 entries)  (d) LLM × Prompt Injection (38 entries)

Figure 11: Technique family evolution over time for four representative concept×topic combinations. Each color represents a technique family; the *y*-axis shows the number of papers per quarter.

**FL-Related Combinations.** The FL-related combinations generally reflect a more mature stage of development, with broader technical diversity and clearer attack-defense co-evolution. As shown in Figure 11a, FL×DP is one of the most technically mature combinations, comprising 697 entries and 18 technique families. Its defense landscape is dominated by anomaly detection (207 papers) and robust aggregation (200 papers), both of which have grown steadily since 2019, while backdoor injection (102 papers) and general data poisoning (92 papers) constitute the two main attack paradigms. Federated secure aggregation (29 papers), including cryptographic approaches such as secure multi-party computation and homomorphic encryption, further represents a smaller but steadily growing defense direction. A similarly mature pattern appears in FL×MIA, which contains 110 entries and 15 technique families. In this combination, membership inference attacks (36 papers) and differential privacy defenses (35 papers) form a near-symmetric pair, suggesting a relatively balanced line of research in which attack and defense have evolved in tandem. Property inference (9 papers) broadens the attack landscape as a related but distinct threat variant.

**LLM-Related Combinations.** In contrast, the LLM-related combinations tend to show rapid expansion but less balanced internal development. As illustrated in Figure 11b, LLM×JB is the largest and most diverse of the LLM settings, with 187 entries and 19 technique families, yet the literature remains heavily concentrated around prompt-level jailbreaks, which alone account for 96 papers and 48% of the total. Defensive work remains comparatively limited, with safety filters (24 papers) being the most common countermeasure, while token-level jailbreaks (16 papers), including GCG-style gradient-based attacks, emerged in late 2023 as a distinct attack paradigm. LLM×DP exhibits a similarly unbalanced structure, though in a different form: backdoor injection dominates the combination with 46 out of 78 entries (59%), whereas general data poisoning (11 entries) and anomaly detection (4 entries) remain much less developed, indicating that defensive work is still in an early stage. Finally, LLM×PI represents a relatively young and narrow area, with only 38 entries and 6 technique families in total. The literature is currently organized around a simple attack-defense structure, with prompt injection attacks (22 papers) opposed mainly by safety filters (9 papers), suggesting that this area is still in an early phase of conceptual and technical expansion.

**Cross-Combination Observations.** Comparing across combinations, we observe three broad maturity patterns. FL×DP represents the most mature and diversified setting, with 18 technique families, a relatively balanced distribution between attack and defense research, and multiple competing defense paradigms. By contrast, LLM×JB appears to be growing but still concentrated: although it also contains 19 technique families, the literature is dominated by a single attack paradigm, namely prompt-level jailbreaks, while defensive work remains comparatively limited. In comparison, combinations such as LLM×PI and T2IM×AE remain nascent, with fewer than 10 technique families and little diversity in defense strategies, suggesting that these areas are still at an early stage of development.

### 4.3 Cross-Concept Shared Technique Families

We analyze which technique families are shared across multiple ML concepts. A technique family is considered "shared" if it appears in at least two ML concepts with three or more unique papers in each.

We identify 17 shared technique families (Figure 12 and Table 6). Six technique families span all five ML concepts: anomaly detection, backdoor injection, general data poisoning, membership inference, adversarial perturbation, and input preprocessing. These results demonstrate that certain attack/defense approaches represent fundamental security concerns that transcend specific ML architectures.

**Technique Family Diffusion Patterns.** We further trace the *temporal adoption order* of shared technique families, defined as the sequence in which a technique family first emerges across different ML concepts. All first-appearance statements below are relative to our corpus, which begins in January 2018, and do not claim absolute historical priority. A consistent pattern can be observed: many technique families appear first in federated learning or graph neural networks, and are later adopted in contrastive learning, LLMs, and text-to-image models. This trend is illustrated by several representative cases. For anomaly detection, the earliest appearances follow FL (2018 Q3) $\rightarrow$ GNN (2019 Q3) $\rightarrow$ LLM (2023 Q3) $\rightarrow$ T2IM (2023 Q3) $\rightarrow$ CL (2023 Q4). Backdoor injection exhibits a similar trajectory, emerging in FL (2019 Q3), then GNN (2020 Q2), CL (2021 Q2), LLM (2021 Q4), and finally T2IM (2022 Q4). Membership inference follows the same overall pattern, first appearing in FL (2018 Q4), then GNN (2020 Q4), CL (2021 Q1), LLM (2022 Q3), and T2IM (2022 Q4).

This FL/GNN-first pattern holds for 7 of the top 10 shared technique families. The exceptions are adversarial perturbation (GNN first), transfer attack (CL first), and watermarking (GNN first). This suggests that FL and GNN, as the earlier and more established ML concepts in security research, serve as early adopters for security technique families that are later explored in newer concepts.

**Attack vs. Defense Paradigm Sharing.** Among the 17 shared technique families, 9 are primarily attack-oriented and 8 are primarily defense-oriented. However, defense technique families tend to have relatively lower cross-concept coverage: 3 defense families (anomaly detection, input preprocessing, adversarial training) span 4 or more concepts, compared to 5 attack families (backdoor injection, data poisoning, membership inference, adversarial perturbation, transfer attack). This suggests that while attack ideas transfer relatively freely across ML concepts, defense solutions may be more concept-specific.

**Why Certain Technique Families Are Shared Across Concepts.** The cross-concept recurrence of certain technique families is not coincidental; it reflects underlying structural commonalities among ML systems. We identify three levels of shareability based on what the paradigm exploits or protects.

The most broadly shared technique families recur across concepts in our corpus, suggesting they are not specific to a single architecture. Data poisoning (5 concepts, 142 unique papers) exploits the universal dependence of ML models on training data: regardless of whether the target is an FL global model, a GNN classifier, or a fine-tuned LLM, corrupting training data can alter model behavior. Membership inference (5 concepts, 90 unique papers) exploits the tendency of models to memorize training data, a property observed across all ML paradigms. Notably, membership inference is the only shared technique family that operates predominantly in a black-box, inference-time setting across *all* concepts (e.g., 0/18 white-box in LLM, 2/15 in T2IM, 2/17 in GNN), indicating that its attack surface is truly independent of model internals. Backdoor injection (5 concepts, 197 unique papers) exploits the fact that any model trained via gradient optimization

Table 5: Top 10 technique families by unique paper count. Two non-technique categories (empirical analysis and toolkit) are excluded. The complete taxonomy of 32 families is in Appendix E.

| Technique Family | #Papers | Atk/Def |
|---|---|---|
| Anomaly detection | 234 | Defense |
| Robust aggregation | 204 | Defense |
| Backdoor injection | 197 | Attack |
| Data poisoning (general) | 142 | Attack |
| Jailbreak (prompt-level) | 98 | Attack |
| Membership inference | 90 | Attack |
| Adversarial training | 79 | Defense |
| Differential privacy | 77 | Defense |
| Adversarial perturbation | 60 | Attack |
| Input preprocessing | 59 | Defense |

Table 6: Top 10 cross-concept shared technique families, ranked by concept coverage. "Papers" counts each unique paper once. "First Appearance" shows the earliest concept-quarter pair.

| Technique Family | #Concepts | #Papers | First Appearance |
|---|---|---|---|
| Anomaly detection | 5 | 234 | FL (2018Q3) |
| Backdoor injection | 5 | 197 | FL (2019Q3) |
| Data poisoning (gen.) | 5 | 142 | FL (2018Q4) |
| Membership inference | 5 | 90 | FL (2018Q4) |
| Adversarial training | 4 | 79 | GNN (2019Q3) |
| Differential privacy | 3 | 77 | FL (2018Q4) |
| Adversarial perturbation | 5 | 60 | GNN (2019Q1) |
| Input preprocessing | 5 | 59 | FL (2021Q2) |
| Watermarking | 4 | 18 | GNN (2021Q4) |
| Transfer attack | 4 | 17 | CL (2022Q4) |

can have hidden behaviors implanted during training, with a highly consistent white-box, training-time threat model across concepts.

A second level of sharing arises from *commonalities in learning paradigms*. Adversarial training (4 concepts, 79 unique papers) applies the min-max optimization framework to any differentiable model, with consistent white-box, training-time threat models across FL, CL, LLM, and GNN. Its absence in T2IM may reflect the difficulty of adapting adversarial training to the denoising process of diffusion models. Transfer attacks (4 concepts, 17 unique papers) exploit shared representation spaces in models that use pre-trained features, with a fully consistent black-box, inference-time threat model.

A third level involves *convergent functional needs*: different concepts independently develop similar techniques to address analogous requirements. Watermarking (4 concepts, 18 unique papers) addresses intellectual property protection across FL, LLM, T2IM, and GNN, but the implementations diverge substantially—FL embeds watermarks in model parameters, while LLM and T2IM embed them in outputs—resulting in inconsistent threat models across concepts.

In contrast, technique families that remain concept-specific typically depend on *architecture-specific features*. Robust aggregation (204 papers) and federated secure aggregation (52 papers) are exclusive to FL because they rely on the distributed client-server aggregation architecture that other concepts lack. Similarly, prompt-level jailbreaks (98 papers), token-level jailbreaks (17 papers), prompt injection (29 papers), and safety filters (35 papers) are exclusive to LLM because they depend on natural language interfaces and instruction-following capabilities absent in FL, CL, and GNN. Among the six concept-specific technique families, three belong to FL and three to LLM, reflecting how these two most-researched concepts have each developed unique security ecosystems that the other cannot leverage.

## 4.4 Implications for ML Security Research

The analyses in this section yield several practical implications. These implications describe where research effort is concentrated or sparse in our corpus. Because they are based on paper counts, they reflect research attention rather than the measured vulnerability of any system, so we present them as suggestions for where to focus future work rather than as assessments of real-world risk. **First, security risks for new ML concepts can be anticipated before attacks emerge.** Technique families that recur across concepts appeared in all five concepts we studied. Any future ML paradigm that relies on training data and gradient optimization will face the same threats. Researchers should begin developing defenses for data poisoning, backdoor injection, and membership inference as soon as a new concept gains traction, rather than waiting for attacks to be demonstrated. **Second, black-box inference-time settings urgently need native defense paradigms.** The current defense ecosystem contains five mature paradigms for white-box training-time settings but only two for black-box inference-time settings (safety filters and input preprocessing). As ML systems move from research prototypes to deployed services, closing this research gap becomes increasingly important. New defense approaches that do not assume access to model parameters or the ability to retrain are worth exploring. **Third, the community should build cross-concept defense**

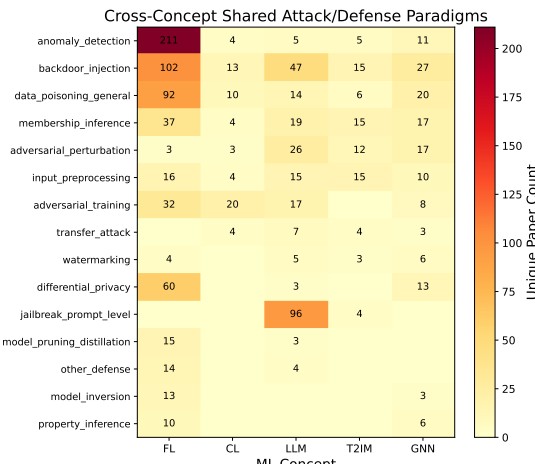

Figure 12: Cross-concept shared technique families. Each cell shows the number of unique papers employing a given technique family within an ML concept. Only families appearing in 2+ concepts with ≥3 papers each are shown.

**infrastructure for universal threats.** Universal attack paradigms are currently met with concept-specific defenses developed in isolation. Shared benchmarks, evaluation protocols, and reusable toolkits would reduce duplicated effort and help defenses mature faster. **Fourth, architecture-specific threats should be identified at design time, not after deployment.** The LLM experience shows that discovering threats only after wide adoption leads to a prolonged defense gap. When designing new architectures, practitioners should proactively analyze what unique attack surfaces the architecture introduces and invest in early defense research.

## 5 Academic Attributions, Collaboration, and Influence Factors (RQ3)

In this section, we investigate the distribution and temporal trend of academic attributions and the corresponding collaboration patterns. Specifically, we study the distribution and temporal evolution of concentration trends in terms of author count, institutes, and regions, and the proportion of different collaboration patterns. Mean and median are two commonly used statistical measures to describe the central tendency of data. The mean reflects the overall trend of the data, while the median is more suitable for describing the typical value, especially when the data distribution is skewed or contains outliers (Khorana et al., 2023). In this section, we primarily use the mean to analyze the overall trend of academic attributions, as the mean considers all data points and intuitively represents the overall level. The results regarding the median will be specifically presented in Section 5.4 to further reveal the typical level of the data and address potential distribution skewness.

### 5.1 Academic Attributions

**Authors.** Figure 13a illustrates the average author count per security paper across five ML concepts. We observed that papers on the topic of LLMs have a significantly higher average author count compared to the other four topics. Specifically, the average author count for LLMs papers exceeds that of the second-ranked contrastive learning by 0.955. Federated learning papers have the fewest average authors; however, the difference was relatively small. Excluding LLMs, federated learning papers have only up to 0.402 fewer authors on average compared to other topics. From a temporal perspective, the average author count per paper has generally increased from 4.000 in 2018 to 5.251 in 2024 (see Figure 26a in Appendix G). However, the average author number of each ML concept does not follow a consistent trend. Notably, since 2021, the average author count for LLMs security papers has always been the highest each year.

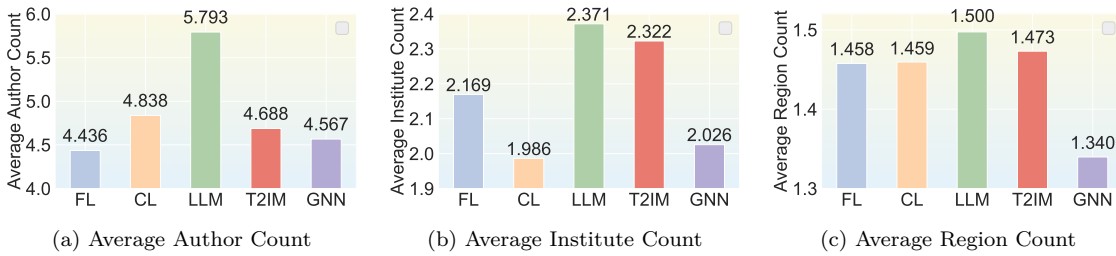

(a) Average Author Count    (b) Average Institute Count    (c) Average Region Count

Figure 13: The Average Author Count, Institute Count, and Region Count for Security Papers of five ML Concepts.

Table 7: Top six regions in terms of number of papers.

| Region | CN | USA | AUS | UK | DE | SGP |
|---|---|---|---|---|---|---|
| # of papers | 739 | 571 | 120 | 109 | 98 | 82 |

**Affiliated Institutes.** In Figure 13b, we present the average affiliated institute numbers of security papers for each ML concept. We observed that similar to the author count, the number of affiliated institutes for security papers related to LLMs is the highest among the five ML concepts, exceeding the lowest (CL) by 0.381. This may be attributed to the nature of LLM research, which often requires more intensive computation and fosters greater collaboration between academia and industry. Additionally, another computationally intensive topic, T2IM, also exhibits a relatively high number of affiliated institutes, ranking second with a mean of 2.322. On the other hand, regarding the temporal change in affiliated institute number, no consistent trend is observed (see Figure 26b in Appendix G). Overall, the number of affiliated institutes remains relatively stable throughout the period we studied.

**Affiliated Regions.** The average numbers of affiliated regions are shown in Figure 13c. The overall results are similar to those observed for institutes, with one notable exception: CL. Although CL has the fewest affiliated institutes (ranking fifth), it does not have the fewest affiliated regions, ranking third in this aspect. Also, we do not observe a consistent temporal trend in the number of regions affiliated with the papers (see Figure 26c in Appendix G).

In addition, we compile the top six regions with the most security papers, as shown in Table 7. China and the United States rank first and second, with 739 and 571 papers, respectively. We observe that ML security research is being widely conducted globally, with the top six regions distributed across three different continents. The word cloud of the abstracts and titles of security papers from the six regions is visualized in Figure 14. We find that "federated learning" is the most prominent term in the security papers from China. On the other hand, the hot spots in the United States are more diverse, with "federated learning," "llm" (including "large language"), "poisoning" (including "backdoor"), and "client" all being popular terms. Security papers focused on LLMs exhibit significantly higher average numbers of authors, affiliated institutions, and regions compared to those on the other four ML concepts. This may suggest that security research on LLMs may attract a broader range of researchers from diverse institutions and countries, highlighting the global interest and collaborative nature of this field.

## 5.2 Collaboration Patterns

We categorize collaboration patterns based on the number of affiliated institutes and regions associated with each paper into three types: independent work, domestic collaboration, and international collaboration. Independent work refers to papers completed either by a single institute or an individual (institute count = 1, region count = 1). Domestic collaboration involves papers jointly completed by multiple institutes, all located within the same region (institute count > 1, region count = 1). International collaboration refers to papers co-authored by multiple institutes, with at least two of them belonging to different regions (institute count > 1, region count > 1).

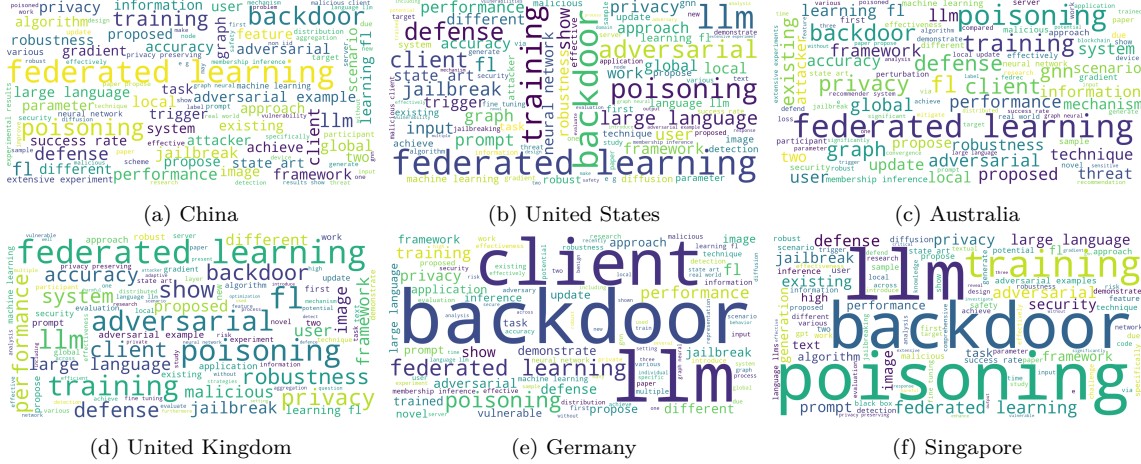

(a) China      (b) United States      (c) Australia

(d) United Kingdom      (e) Germany      (f) Singapore

Figure 14: The word clouds for the top six regions with ML security papers.

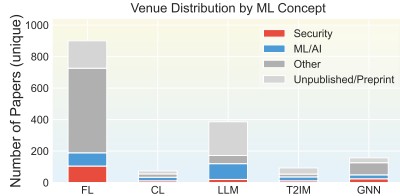

Figure 15: Publication venue distribution by ML concept. Venues are classified into security conferences/journals, ML/AI conferences/journals, other venues, and unpublished preprints.

The nested pie chart in Figure 27 in Appendix G illustrates the proportion of papers corresponding to the three collaboration patterns. Specifically, 545 papers were completed independently, accounting for 34.4% of the total. The number of papers involving domestic collaboration is 529, representing 33.4% of the total, while those resulting from international collaboration are 510, making up 32.2%. The distribution of papers across the three collaboration patterns is relatively balanced.

The number of papers completed through independent efforts, domestic collaboration, and international collaboration is evenly distributed. This shows that the research collaboration patterns in this field are flexible, and the field demonstrates both openness and a globalized nature.

## 5.3 Publication Venue Distribution

We classify publication venues into four categories: security venues (e.g., S&P, CCS, USENIX Security, NDSS, TIFS), ML/AI venues (e.g., NeurIPS, ICML, ICLR, CVPR, ACL), other venues, and unpublished/preprints (primarily arXiv).

The venue distribution varies substantially across ML concepts. LLM security papers have the highest unpublished rate (55.4%) and the lowest security venue rate (5.2%), consistent with the preprint-first publication culture in the LLM community. In contrast, FL and GNN security papers have lower unpublished rates (19.3% and 19.7%, respectively) and are more likely to appear in security venues (11.7% and 15.3%). CL security papers have the highest ML/AI venue rate (31.1%), reflecting the closer alignment of contrastive learning with the ML community.

These venue patterns, shown in Figure 15, suggest that ML security research straddles two communities—security and ML—with the balance varying by ML concept. LLM security research is more strongly anchored in the ML community, while FL and GNN security research has deeper roots in the security community. The full venue classification rules are provided in Appendix E.

### 5.4 Potential Factors Affecting Academic Influence

The academic influence of a paper refers to the extent to which the paper has had an impact on the academic community, which can be affected by a combination of factors. In this section, we investigate whether the academic influence is associated with six different factors, i.e., ML concepts, security topics, author count, regions, collaboration patterns, and publication status, through hypothesis testing.

#### 5.4.1 Experimental Settings

**Academic Influence.** The academic influence of a paper is defined as the degree to which it has impacted the academic community. This influence can be quantified using two metrics: citation count and citation density (i.e., the average number of citations per day since the paper's publication). Previous studies have demonstrated that both metrics serve as effective indicators of a publication's academic influence (Jones et al., 2017; Kadic et al., 2020; Sandison, 1975). Given that citation density removes the effects of time, we employ this metric to assess the academic influence of a paper.

**Key Factors.** Through hypothesis testing, we explore the relationship between academic influence (i.e., citation density) and six key factors: ML concepts, security topics, author count, geographic regions, collaboration patterns, and publication status. Furthermore, we investigate the specific conditions within each factor that correlate with stronger academic influence.

**Sample Size.** For the hypothesis testing in this section, we analyze at the unique paper level rather than the entry level used in Sections 3–4. Since some papers appear in multiple concept-topic combinations, we deduplicate by paper ID. Of the 1,591 unique papers, 1,569 have sufficient metadata (publication date and citation count) to compute citation density; the remaining 22 papers are excluded from the hypothesis testing due to missing dates. As shown in the supplementary tables in Appendix G, the sample sizes for the ML concept ($x_1$) and security topic ($x_2$) analyses are 1,569 (sum of all group counts). For other variables ($x_3$–$x_6$), sample sizes range from 1,475 to 1,569 due to additional missing metadata in some fields.

#### 5.4.2 Statistical Methodology

**Overview.** In this section, we perform comprehensive statistical tests to explore the relationships between academic influence, measured as citation density (dependent variable), and six independent variables. Details of the statistical tests conducted in this study are presented in Appendix C. For each test, we begin by examining the citation density distributions within each group of the independent variables. Based on the characteristics of these distributions, we select appropriate hypothesis tests to assess whether statistically significant differences exist among the groups. Subsequently, we conduct post-hoc analyses to derive fine-grained insights. We aim to assess if the specific group pairs exhibit statistically significant differences and understand the extent of these differences in citation density distributions. In the following, we provide a detailed explanation of each step in our assessment.

**Variables of Statistical Tests.** We use the following variable settings for our statistical tests:

- **Dependent Variables:** $y$ = citation density.

- **Independent Variables:** $x_1$ = ML concepts, $x_2$ = security topics, $x_3$ = author count, $x_4$ = regions, $x_5$ = collaboration patterns, and $x_6$ = publication status.

**Distributional Tests.** In this study, for each independent variable, we perform a one-sample Kolmogorov-Smirnov test on each group to verify whether the data follows a normal distribution. The test compares the empirical cumulative distribution function (ECDF) of the sample with the cumulative distribution function (CDF) of a theoretical normal distribution, and determines whether the data follows a normal distribution. The null hypothesis assumes that the data follows a normal distribution. We use the $p$-value to determine whether the data within each group of the independent variable follows a normal distribution. If the $p$-value is less than the significance level ($\alpha = 0.05$), the null hypothesis is rejected, indicating that the data in that group does not follow a normal distribution. The results of this test guide the selection of the appropriate subsequent statistical analysis.

**Overall Difference Test.** We use hypothesis testing to examine whether there are statistically significant differences in citation density between the different groups of an independent variable, thereby assessing whether there is a statistical relationship between the independent and dependent variables. Based on the results of the normality test, we determine whether to use parametric or non-parametric tests for hypothesis testing. Since the dependent variable is numerical, if the normality test indicates that each group follows a normal distribution, we will use parametric tests, such as the t-test (Kim, 2015) for comparing two groups or ANOVA (Scheffé, 1999) for comparing more than two groups. If the normality assumption is violated, we will use non-parametric tests, such as the Mann-Whitney U test (McKnight & Najab, 2010; Nachar et al., 2008) for comparing two independent groups or the Kruskal-Wallis H test (McKight & Najab, 2010; Ostertagová et al., 2014) for comparing three or more groups. The hypotheses are formulated as follows:

$$H_{0i} : \text{The distribution of } y \text{ is identical across groups of } x_i.$$
$$H_{1i} : \text{The distribution of } y \text{ varies by different groups of } x_i.$$

where $i \in \{1, \ldots, 6\}$. We use the $p$-value to determine whether the data within each group of the independent variable follows a normal distribution. If the $p$-value is less than the significance level ($\alpha = 0.05$), the null hypothesis is rejected, indicating that there are statistically significant differences in the distribution y between the different groups of $x_i$.

**Post-Hoc Tests.** After conducting an overall difference test, such as the Kruskal-Wallis H test, we use Dunn's test (Dinno, 2015; Pohlert, 2014) for post-hoc analysis to further investigate which specific groups exhibit statistically significant differences. The null hypothesis of Dunn's test assumes that the two groups being compared have the same distribution. Dunn's test provides a $p$-value for each pairwise comparison. To reduce the Type I error rate (Banerjee et al., 2009), we apply Bonferroni correction (Armstrong, 2014) to the $p$-values to determine whether there are statistically significant differences between the groups. While Bonferroni correction primarily addresses the risk of false positives, it can also provide a more reliable framework for interpreting results when sample sizes are unbalanced. If the $p$-value is less than the significance level ($\alpha = 0.05$), the null hypothesis is rejected, indicating that there are statistically significant differences in y between these two groups of $x_i$. Afterwards, by combining the results of Dunn's test and analyzing the median for each group presented in the boxplot of $x_i$, we can determine under what specific conditions certain categories tend to exhibit higher citation density, complementing the statistical findings from Dunn's test.

**Why Focus on Medians.** The median is a robust measure of central tendency that is less influenced by outliers and variability. This makes it a more reliable measure when data is not symmetrically distributed (e.g., skewed publication distributions) or contains extreme values (e.g., highly cited publications) (Khorana et al., 2023). In such cases, the median provides a more accurate representation of the typical value within a dataset.

### 5.4.3   Results

The distribution test results show that the $p$-values for all groups across all independent variables are less than the significance level. The highest $p$-value, 0.049, is observed for the group with the author count of 1 under the independent variable $x_3$, which is still below 0.05. This indicates that the citation density distributions for all groups of all independent variables do not follow a normal distribution. Consequently, we employ non-parametric tests for the next step, the overall difference test. Depending on the number of groups within each independent variable, we use either the Mann-Whitney U test or the Kruskal-Wallis H test.

**ML Concept.** For the variable $x_1$, which includes more than two groups, the Kruskal-Wallis H test is performed to assess overall differences. The calculated $p$-value is less than 0.001, which is significantly lower than the threshold for statistical significance ($\alpha = 0.05$), leading to the rejection of the null hypothesis. The distribution of citation density differs statistically significantly across different ML concepts. This shows that the citation density of a paper is related to its ML concepts.

Further analysis using Dunn's test, combined with the statistics presented in the boxplot, allows for a more detailed examination of which specific groups within $x_1$ exhibit higher citation density. Each cell of the

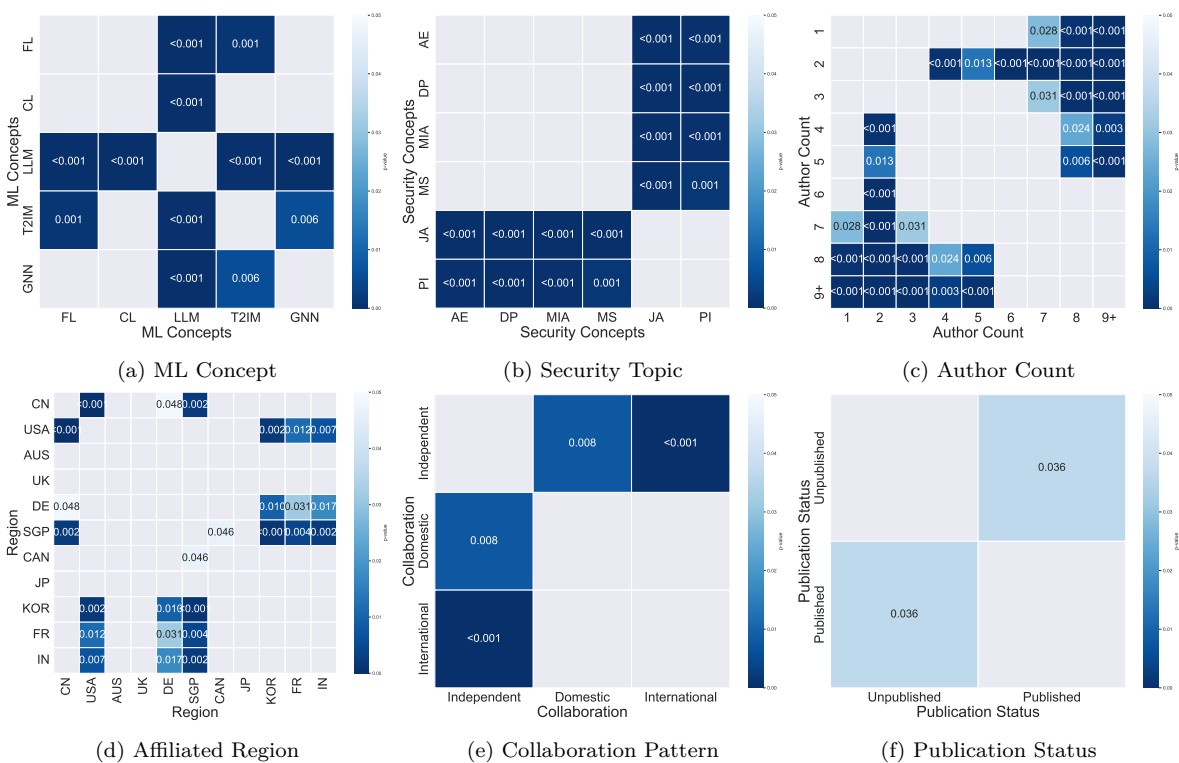

Figure 16: *p*-value metrics of Dunn's test for citation density by different independent variables. Those with *p*-values larger than 0.05 are omitted.

*p*-value matrix in Figure 16 shows the *p*-value obtained from Dunn's test between two groups, and for ease of reading, we only show cells with a *p*-value less than 0.05. The results from Figure 16a demonstrate that the citation density of papers grouped under LLM differs statistically significantly from all other groups. Considering the median values shown in Figure 17a (with detailed statistics provided in Table 14 in Appendix G), it can be concluded that papers associated with LLM tend to exhibit higher citation densities compared to other ML concepts.

Security papers addressing LLMs are likely to have a greater academic impact than those focused on other machine learning concepts. This indicates that, in the security domain, researchers show greater interest in LLMs than in the other four categories of ML concepts. Similarly, the analysis of security topics reveals that researchers pay more attention to data poisoning and prompt injection than to the other four categories. Notably, prompt injection is an attack specifically targeting LLMs. This further confirms the statistically significant attention that LLMs have received in the security domain.

**Security Topic.** Regarding the overall difference test, the *p*-value of the Kruskal-Wallis H test on $x_2$ is below 0.001, which is well below the statistical significance level ($\alpha = 0.05$). Thus, the null hypothesis is rejected. This indicates that the security topics of a paper may affect its citation density.

The results of Dunn's test indicate statistically significant differences in citation density distributions for papers on jailbreak when compared to those on adversarial examples, jailbreak attacks, membership inference attacks, and model stealing, as shown in Figure 16b. Similarly, papers on prompt injection also exhibit statistically significant differences in citation density compared to the same four security topics. Moreover, the median values presented in Figure 17b (with detailed statistics provided in Table 16 in Appendix G) further support this finding. Consequently, we conclude that papers addressing security topics like jailbreak and prompt injection tend to have higher citation densities than those on other topics.

**Author Count.** When categorizing the paper data based on the author count, we find that papers with more than 9 authors are rare. For example, only six papers have 11 authors, and just one paper has 12

authors. To avoid sparse data and to ensure more balanced sample sizes across the groups, we combine papers with 9 or more authors into the same group, labeling them as "9+ authors."

The Kruskal-Wallis H test for overall differences on $x_3$ yields a $p$-value below 0.001, well below the statistical significance level ($\alpha = 0.05$). The null hypothesis is rejected. This suggests that the citation density on a paper is associated with its author count.

Regarding the post-hoc test on the author count, the $p$-value metrics in Figure 16c, combined with the median information in Figure 17c (detailed statistics can be found in Table 17 in Appendix G), indicate that papers with more than 8 authors tend to have higher citation densities compared to papers with fewer than 5 authors.

Papers with more than eight authors tend to exhibit stronger academic influence compared to those with fewer than five authors. A higher number of authors increases the likelihood of the paper being a product of cross-institutional or interdisciplinary collaboration, potentially broadening its promotional reach.

**Affiliated Region.** In order to prevent the data from being too sparse as well as to ensure the credibility of the statistical results, here we only study regions with more than 30 papers. There are eleven regions with more than 30 papers, in descending order of the number of papers, namely China, the United States, Australia, the United Kingdom, Germany, Singapore, Canada, Japan, South Korea, France, and India.

The Kruskal-Wallis H test for overall differences on $x_4$ produces a $p$-value below 0.001, which is significantly lower than the statistical significance ($\alpha = 0.05$). Therefore, the null hypothesis is rejected, indicating that the affiliated region of a paper is associated with its citation density.

The post-hoc test on the affiliated regions, as shown in the $p$-value metrics in Figure 16d and the median values in Figure 17d (detailed statistics is presented in Table 18 in Appendix G), reveals that the papers affiliated with Singapore tend to have higher citation densities compared to those affiliated with China, Canada, South Korea, France, and India. However, no evidence suggests that papers from any specific region exhibit a statistically significantly higher citation density compared to the citation density of all other regions.

**Collaboration Pattern.** The Kruskal-Wallis H test for general differences in $x_5$ produces a $p$-value below 0.001, significantly lower than the statistical significance level ($\alpha = 0.05$), leading to the rejection of the null hypothesis. The result shows that the collaboration pattern probably influences its citation density. Further analysis of the $p$-value metrics for collaboration patterns from Dunn's test is shown in Figure 16e, along with the median information in Figure 17e (comprehensive statistical data is provided in Table 19 in Appendix G. This reveals that papers co-authored tend to have higher citation densities compared to those completed independently.

Collaborative papers generally have greater academic influence than those completed independently. Cross-institutional collaborations benefit from broader dissemination channels through the networks of participating institutes, thereby enhancing the academic influence of the paper. To improve the academic influence of their work, authors could consider engaging in cross-institutional collaborations.

**Publication Status.** In $x_6$, the data is divided into two groups: published and unpublished. We used the Mann-Whitney U test to conduct an overall difference test to assess whether there is a statistically significant difference between the distributions of the two independent samples. The $p$-value is 0.036, which is below the statistical significance level ($\alpha = 0.05$), suggesting that the publication status of papers likely influences their citation density.

In the univariate test, unpublished papers show a slightly higher median citation density (Figure 17f, with detailed statistics in Table 20 in Appendix G). This univariate comparison is, however, confounded by publication age and topic popularity, because unpublished preprints in our corpus are disproportionately recent and concentrated in fast-moving, high-citation areas such as LLM security. To control for these factors, we fit a negative-binomial regression of citation count with log(days since publication) as an offset and ML concept, security topic, author count, region, and publication status as predictors (Appendix C). After this multivariable control, the apparent preprint advantage reverses: unpublished status is associated with about 23% *lower* citation density (incidence rate ratio = 0.77, 95% CI [0.65, 0.92], $p = 0.004$). We

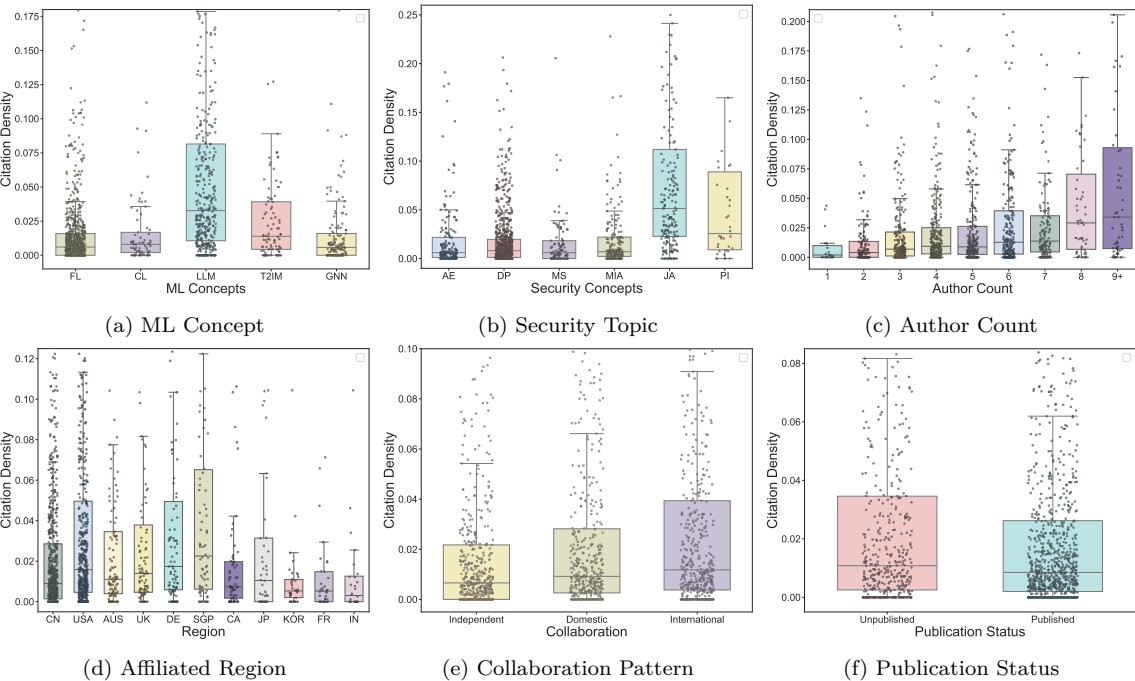

Figure 17: Boxplot for citation density by different independent variables.

therefore do not claim a citation advantage for preprints; the univariate difference is explained by the concentration of preprints in high-citation concept and topic areas.

### 5.5 Summary

All six factors we examined are statistically significantly associated with citation density. Three observations stand out for their practical relevance. **First, concept and topic selection are the dominant correlates of citation density.** In the multivariable model (Appendix C), LLM papers (incidence rate ratio 3.93) and jailbreak papers (incidence rate ratio 3.34) have several times the citation density of the reference categories, and these effects remain strong after controlling for time, author count, region, and publication status. Researchers seeking impact should consider aligning their work with high-attention areas, while also recognizing that under-explored areas offer less competition and potentially higher marginal contribution. **Second, collaboration correlates with influence.** Papers with more authors and cross-institutional or international collaboration tend to receive more citations. While causality cannot be established from our data, this pattern is consistent with the view that diverse teams produce work with broader reach and visibility. **Third, preprints do not have an intrinsic citation advantage.** Although unpublished papers show a slightly higher citation density in the univariate comparison, this difference reverses after multivariable control (incidence rate ratio 0.77, $p = 0.004$): it is explained by the concentration of preprints in high-citation areas such as LLM security, not by an intrinsic advantage of preprint dissemination.

## 6 Discussion

**Defense Gaps as Research Opportunities.** Our attack-defense imbalance analysis shows that defense research is sparsest relative to attack research in LLM security, where jailbreak defense ratio is only 0.30 and membership inference defense ratio is 0.26. Similarly, text-to-image model security (T2IM×MIA = 0.20) and GNN security (GNN×DP = 0.30) present significant defense gaps. These imbalances describe the state of the field through June 2024 and indicate where defense effort was most needed as of that snapshot. For instance, while 145 papers have proposed jailbreak attacks against LLMs, only 60 have

proposed corresponding defenses, suggesting that the community may benefit from shifting focus toward defense development in these areas.

**Cross-Concept Transfer of Security Techniques.** Our temporal adoption analysis reveals a consistent pattern: within our corpus (from January 2018 onward), security techniques first appear in federated learning and graph neural networks, and are subsequently explored in LLMs and text-to-image models. This pattern has practical implications. For researchers working on newer ML concepts, techniques developed for earlier paradigms may serve as a starting point. For example, anomaly detection methods for FL could inspire novel defenses against LLM data poisoning. However, the effectiveness of such cross-concept transfer is not guaranteed—the threat models, data modalities, and system architectures differ substantially across concepts. Validating the transferability of defense techniques across ML paradigms is an important open problem.

**The LLM Security Paradox.** LLM security research exhibits a striking combination of characteristics: the highest paper volume growth, the lowest publication rate (with 55% of papers remaining unpublished), the most severe attack-defense imbalance, and the highest citation density. This suggests a field that is simultaneously the most active, the most impact-generating, and the least settled in its publication and defense research patterns. The preprint-dominated publication pattern is consistent with the hypothesis that findings in this area may become outdated quickly, as vulnerabilities are frequently patched before papers complete the peer-review cycle, though other explanations (e.g., the novelty of the field, different community norms) are also plausible.

**Influence Enhancement.** Our findings suggest that papers involving collaborations across multiple regions, institutions, and authors tend to exhibit higher academic impact. This may be attributed to the diversity of perspectives and feedback introduced through collaboration, which can enhance both the quality and influence of the work.

**Developments Since the Data Cutoff.** Our corpus ends on June 30, 2024. Because the field moves quickly, the specific numbers in this paper should be read as a mid-2024 snapshot rather than a current-day map. We summarize the post-cutoff shifts that would most update our quantitative conclusions, based on a non-systematic reading of the literature after the cutoff. First, several attack families that are absent or rare in our taxonomy have since grown, including agentic and tool-use attacks (Debenedetti et al., 2024), retrieval-augmented-generation and memory poisoning (Zou et al., 2025; Chen et al., 2024), fine-tuning-API attacks (Qi et al., 2024), sleeper-agent and alignment-faking behavior (Hubinger et al., 2024), and many-shot and best-of-$N$ jailbreaks (Anil et al., 2024; Hughes et al., 2024). Second, the LLM jailbreak defense ratio of 0.30 is a mid-2024 figure and likely understates 2025 and 2026 defense activity, given newer directions such as circuit breakers (Zou et al., 2024), deliberative alignment (Guan et al., 2024), robust refusal training such as R2D2 (Mazeika et al., 2024), and latent-space adversarial training (Casper et al., 2024). Third, Section 3.4 already shows LLM volume overtaking federated learning in early 2024, so our federated-learning-centric framing over-represents the earlier part of the window. Fourth, the citation-density results in Section 5.4 are sensitive to additional years of accumulation and should be read as describing the 2018 to mid-2024 window. The structural observations of this paper, namely the white-box-training versus black-box-inference contrast, the concept-independent shared paradigms, and the federated-learning and GNN early-adopter pattern, are largely time invariant and are not affected by this gap.

# 7 Limitations & Future Work

In our study, we made every effort to minimize bias and identify limitations. However, we acknowledge that certain biases and limitations remain unavoidable. First, we collect the original paper dataset by performing relevant searches on the titles and abstracts of the papers with different keywords through the Semantic Scholar API. While we strive to expand the set of keywords for each topic as thoroughly as possible, a small number of less relevant yet related terms may be inadvertently overlooked. This could result in the omission of a few security papers from the original dataset. A single-year recall check for our single-keyword sets (Appendix A) suggests this omission is small, but it is estimated rather than exhaustive. Second, to ensure the high quality of the final cleaned dataset, we manually annotate the original dataset to identify security papers genuinely relevant to our research topics. However, we also recognize the inherent subjectivity

of manual judgment (e.g., different experts may have conflicting opinions on whether a paper should be included). To minimize bias caused by subjectivity, we make each piece of data judged by at least 3 or more experts, and the majority opinion prevails. Third, our findings are limited to the security papers released between January 1, 2018 and June 30, 2024. As new security papers continue to emerge, they may not align with the patterns identified in our study, and we therefore present our results as a baseline snapshot rather than a current-day map, with post-cutoff developments discussed in Section 6. We leave the analysis of these papers for future research. Fourth, our annotation pipeline relies on LLM-assisted labeling validated both by in-sample human review and by an out-of-sample human re-annotation of a held-out 100-paper sample (Appendix E.1). Extending this out-of-sample validation to a larger, concept×topic sample and reporting a full per-family confusion matrix would further strengthen it and is a direction for future work. Fifth, the defense ratio and defense lag are based on paper counts, so they measure the relative allocation of research effort rather than the empirical robustness of deployed systems.

## 8 Related Work

**ML Security Surveys.** Numerous surveys have been published on specific ML security topics. In adversarial ML, previous studies provide comprehensive overviews of adversarial attacks and defenses (Biggio & Roli, 2018; Chakraborty et al., 2021; Guan et al., 2018; Chowdhury et al., 2024). For federated learning security, prior surveys examine privacy and security challenges in distributed settings (Kairouz et al., 2021; Kuntla et al., 2021). LLM safety has been surveyed by recent works focusing on jailbreaks, alignment, and prompt injection (Rosenberg et al., 2021; Paracha et al., 2024). However, these surveys are typically scoped to a single ML concept or security topic and rely on narrative synthesis rather than systematic quantitative analysis. Our work differs in two key aspects: (1) we cover *all* combinations of five ML concepts and six security topics simultaneously, enabling cross-concept comparisons that single-topic surveys cannot provide; and (2) we employ quantitative methods (LLM-assisted annotation, statistical hypothesis testing) rather than narrative review, enabling reproducible and scalable analysis.

**Meta-Studies of Academic Research.** In the broader academic community, meta-studies have been used to understand publication patterns and research dynamics. Prior work emphasizes the importance of meta-studies for improving science policy (Fortunato et al., 2018). Other work focuses on the over-optimization of academic publishing metrics (Fire & Guestrin, 2018). Prior studies find that international collaboration correlates with higher citations (Smart & Bayer, 1986; Wang et al., 2024). In computer science specifically, meta-studies have examined artifact availability (of Sciences Engineering et al., 2019; Vandewalle et al., 2009; Raghupathi et al., 2022; Collberg & Proebsting, 2016), code quality (Trisovic et al., 2021), and reproducibility in ML security (Olszewski et al., 2023; Hamm et al., 2019). Our work extends this tradition by combining bibliometric analysis with technique-level annotation, bridging the gap between high-level publication statistics and the technical content of the research.

## 9 Conclusion

In this study, we conducted a data-driven survey of 1,591 papers in the ML security domain, going beyond traditional bibliometric analysis to examine attack-defense dynamics, technical evolution, and cross-concept patterns. Our findings reveal three key contributions. First, we quantify a systematic attack-defense imbalance: defense research significantly lags behind attack research in emerging areas such as LLM jailbreaks and text-to-image model security, while mature areas like federated learning have achieved greater balance. Second, using LLM-assisted annotation of all paper abstracts, we identify 32 technique families and trace their evolution over time, revealing that LLM security remains dominated by a single attack paradigm (prompt-level jailbreaks) with limited defense diversity. Third, we identify 17 technique families shared across multiple ML concepts, with a consistent temporal adoption pattern from federated learning to newer concepts such as LLMs and text-to-image models. Additionally, we confirm statistically significant associations between academic influence and six factors, including ML concepts, security topics, author count, regions, collaboration patterns, and publication status. We hope that our findings, particularly the defense gap analysis and cross-concept paradigm mapping, will help researchers identify high-impact research directions in ML

security. As the field is developing quickly, we present these findings as a January 2018 to June 2024 baseline intended for longitudinal comparison rather than as a present-day map of the field.

## 10    Ethical Considerations

We adhere to ethical guidelines and strict data privacy standards, analyzing only publicly available information, such as citation data, publication venues, available dates, and affiliated institutes. Proprietary or sensitive data, as well as harmful data and personal information about authors or developers, are strictly excluded from our analysis.

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

# A  Keyword Set Selection Process

The keywords for each keyword set were determined through a rigorous and systematic process. Specifically, five experts in the ML security field conducted independent evaluations and engaged in extensive discussions, ultimately reaching a consensus. The final keyword sets of the ML concepts and the security topics are shown in Table 8 and Table 9, respectively. The sets of keyword pairs to collect security papers of a security topic against an ML concept are then constructed as follows:

```
[A keyword set of an ML concept] + [A keyword set of a security topic]
```

Table 8: Keyword sets of ML concepts.

| ML Concept | Keyword Set |
|---|---|
| FL | [federated learning, federated reinforcement learning, federated transfer learning] |
| CL | [contrastive learning] |
| LLM | [large language models, multimodal models, in-context learning, prompt-based learning, vision language models, multimodal large language models] |
| T2IM | [text-to-image models, diffusion models, stable diffusion, diffusion-based models, latent diffusion] |
| GNN | [graph neural networks, graph convolutional networks, graph attention networks, variational graph autoencoders, subgraph neural networks] |

**Keyword Recall Check.** Three of our sets use a single keyword (contrastive learning, jailbreak, prompt injection), so we estimate their recall following a single-year sensitivity analysis. For the year 2023, we

Table 9: Keyword sets of security topics.

| Security Topic | Keyword Set |
|---|---|
| AE | [adversarial examples, adversarial machine learning, adversarial learning] |
| DP | [data poisoning, poisoning attacks, trojan attacks, backdoor attack] |
| MS | [model stealing, model extraction, model piracy, knowledge extraction, model inversion, model-stealing, hyper-parameter stealing, hyperparameter stealing, parameter stealing] |
| MIA | [membership inference attacks, property inference attacks, attribute inference attacks, inference attacks] |
| JB | [jailbreak] |
| PI | [prompt injection] |

query the Semantic Scholar API with the original keyword and with an expanded synonym set within a security context, and compare candidate counts. For contrastive learning, we add SimCLR, MoCo, self-supervised contrastive, InfoNCE, and contrastive representation; for jailbreak, we add jailbreaking, safety bypass, refusal bypass, and prompt-based attack; for prompt injection, we add indirect prompt injection, instruction injection, and goal hijacking. The expanded sets change the raw candidate pool only marginally: contrastive learning $236 \rightarrow 243$ $(+3\%)$, jailbreak $69 \rightarrow 69$ $(+0\%)$, and prompt injection $27 \rightarrow 28$ $(+4\%)$. Since the expanded query is a superset of the single keyword, we manually adjudicated the difference set, that is, the papers found only by the expanded synonyms. Of the seven additional contrastive-learning candidates, none were in scope (they concern image-to-image translation, imitation learning, robotic grasping, and medical imaging); the jailbreak difference set was empty; and the single additional prompt-injection candidate was in scope (a multi-modal indirect-injection paper phrased as "indirect prompt and instruction injection"). The single keywords, therefore, missed 0, 0, and 1 in-scope papers respectively in 2023, indicating high recall within the keyword-defined candidate pool.

**Scope and Definitions of Concepts and Topics.** We selected the five ML concepts and six security topics because they are among the most prominent and widely studied in recent years, and fall within the domain expertise of our annotators. We do not claim that these lists are exhaustive or that the categories are mutually exclusive; our choice follows existing taxonomies of ML security (Chowdhury et al., 2024; Rosenberg et al., 2021; Paracha et al., 2024). To keep the categories as separable as possible, we use the following operational definitions. *Adversarial examples (AE)*: test-time input perturbations that cause an incorrect output without changing the model or its training data. *Data poisoning (DP)*: manipulation of training data or the training process, including backdoor and trojan triggers, to corrupt the trained model. *Model stealing (MS)*: recovering a model's parameters, architecture, or functionality through query access. *Membership inference (MIA)*: inferring properties of the training data, such as whether a sample was used in training, from access to the model. *Jailbreak (JB)*: crafted inputs that bypass the safety alignment of a generative model to elicit disallowed content. *Prompt injection (PI)*: adversarial instructions delivered through untrusted content or data so that the model follows the attacker rather than the user. We also observe that some papers touch more than one topic, for example, a jailbreak implemented through an optimized adversarial suffix. In such cases, the topic is assigned by the keyword pair used to retrieve the paper and then confirmed by human annotators using majority vote, consistent with the overall inter-annotator agreement reported in Appendix E.

# B    Data Sources

Previous studies have indicated that Semantic Scholar and Google Scholar have similar indexing coverage in the computer science field (Hannousse, 2021). However, due to the lack of a reliable official API for Google Scholar, we ultimately chose to use Semantic Scholar to obtain the original dataset. Semantic Scholar (Semantic Scholar, 2024a) is a free and powerful academic search engine containing over 200 million papers from all scientific fields and covering papers from different publishers and preprint databases. In this study, we used the Semantic Scholar API to collect metadata for various papers and their corresponding information, including attributes such as citation count, publication venues, authors, and so on.

However, while obtaining data from Semantic Scholar, we found that some data were missing certain attributes, such as publication venue. To address this, we used DBLP to retrieve the missing attributes. DBLP (DBLP, 2024) is a well-known online database that provides bibliographic information for major computer science publications. It includes a large number of research papers, conference proceedings, and journal papers from computer science and related fields. DBLP also indexes works from many publishers and preprint databases and provides detailed information about authors, titles, publication venues, and other relevant metadata.

## C   Details of the Statistical Tests in Our Studies

**The Kolmogorov-Smirnov (KS) Test.** is a widely employed non-parametric statistical test used to evaluate whether a sample is drawn from a specific distribution, with its most common application being the assessment of normality (Berger & Zhou, 2014; Lilliefors, 1967).

**The Mann-Whitney U Test.** is a non-parametric test used to assess whether there are statistically significant differences between the distributions of two independent samples, particularly when the data does not follow a normal distribution.

**The Kruskal-Wallis H Test.** is a non-parametric test used to assess whether there are statistically significant differences in the distributions of three or more independent samples. It is an extension of the Mann-Whitney U test, designed to handle differences between multiple groups, and does not require the assumption of normality in the data.

**Dunn's Test.** is a non-parametric post-hoc test used for pairwise comparisons between multiple groups after a statistically significant result from the Kruskal-Wallis H test (Ruxton & Beauchamp, 2008). It helps identify which specific groups differ from each other when there is evidence that at least one group's distribution is different by comparing the mean ranks of the groups.

**Defense-Ratio Robustness.** To check that the defense-ratio rankings do not depend on how dual attack-and-defense ("Both") papers are counted, we recompute the per-cell defense ratio under three rules: (i) excluding "Both" papers, (ii) counting them as defense (the rule used in the main text), and (iii) counting them as attack. Only 37 of 1,682 entries are labeled "Both," so each ratio changes by at most 0.05. Table 10 reports all combinations with at least 20 entries; the set of most under-defended combinations is identical under all three rules.

Table 10: Defense-ratio sensitivity to the "Both" counting rule. $a$, $d$, $b$ are attack-only, defense-only, and Both entry counts. Combinations with at least 20 entries are shown.

| Combination | $n$ | $a$ | $d$ | $b$ | (i) exclude | (ii) Both=def | (iii) Both=atk |
|---|---|---|---|---|---|---|---|
| FL×AE | 48 | 11 | 36 | 1 | 0.766 | 0.771 | 0.750 |
| FL×DP | 706 | 187 | 499 | 20 | 0.727 | 0.735 | 0.707 |
| FL×MS | 76 | 10 | 66 | 0 | 0.868 | 0.868 | 0.868 |
| FL×MIA | 111 | 48 | 61 | 2 | 0.560 | 0.568 | 0.550 |
| CL×AE | 30 | 9 | 21 | 0 | 0.700 | 0.700 | 0.700 |
| CL×DP | 35 | 25 | 10 | 0 | 0.286 | 0.286 | 0.286 |
| LLM×AE | 55 | 38 | 16 | 1 | 0.296 | 0.309 | 0.291 |
| LLM×DP | 82 | 57 | 24 | 1 | 0.296 | 0.305 | 0.293 |
| LLM×MIA | 23 | 17 | 6 | 0 | 0.261 | 0.261 | 0.261 |
| LLM×JB | 200 | 140 | 55 | 5 | 0.282 | 0.300 | 0.275 |
| LLM×PI | 40 | 25 | 13 | 2 | 0.342 | 0.375 | 0.325 |
| T2IM×AE | 38 | 17 | 21 | 0 | 0.553 | 0.553 | 0.553 |
| T2IM×DP | 29 | 21 | 8 | 0 | 0.276 | 0.276 | 0.276 |
| T2IM×MIA | 20 | 16 | 3 | 1 | 0.158 | 0.200 | 0.150 |
| GNN×AE | 33 | 21 | 11 | 1 | 0.344 | 0.364 | 0.333 |
| GNN×DP | 64 | 45 | 18 | 1 | 0.286 | 0.297 | 0.281 |
| GNN×MIA | 45 | 20 | 24 | 1 | 0.545 | 0.556 | 0.533 |

**Multivariable Citation-Density Model.** To isolate the effect of publication status from confounders, we fit a negative-binomial regression of citation count with log(days since publication) as an offset, so the coefficients describe citation density (citations per day). The model uses $n = 1,477$ papers (those with a citation count, publication date, authors, and a valid concept and topic). Since the multivariable model applies

*listwise* deletion across all predictors, this is slightly smaller than the 1,569 papers used in the univariate tests of Section 5.4. Predictors are ML concept (reference FL), security topic (reference AE), publication status (reference Published), author count, and number of regions. The Poisson Pearson dispersion is 68.5, confirming overdispersion and justifying the negative binomial. Coefficients are reported as incidence rate ratios (IRR), the multiplicative effect on citation density (Table 11). After multivariable control, unpublished status is associated with lower citation density (IRR 0.77), reversing the univariate result, while concept and topic effects remain large.

Table 11: Negative-binomial GLM of citation count with $\log$(days since publication) offset ($n = 1,477$). IRR is the multiplicative effect on citation density.

| Predictor (reference) | IRR | 95% CI | $p$ |
|---|---|---|---|
| LLM (vs. FL) | 3.93 | [2.99, 5.18] | $< 0.001$ |
| T2IM (vs. FL) | 1.92 | [1.38, 2.68] | $< 0.001$ |
| CL (vs. FL) | 1.08 | [0.76, 1.54] | 0.68 |
| GNN (vs. FL) | 0.88 | [0.68, 1.14] | 0.33 |
| Jailbreak (vs. AE) | 3.34 | [2.36, 4.72] | $< 0.001$ |
| Data poisoning (vs. AE) | 1.34 | [1.04, 1.73] | 0.025 |
| MIA (vs. AE) | 1.33 | [0.98, 1.80] | 0.067 |
| Model stealing (vs. AE) | 1.05 | [0.73, 1.51] | 0.80 |
| Prompt injection (vs. AE) | 1.49 | [0.86, 2.57] | 0.15 |
| Unpublished (vs. Published) | 0.77 | [0.65, 0.92] | 0.004 |
| Author count (per author) | 1.04 | [1.00, 1.08] | 0.050 |
| Number of regions (per region) | 1.28 | [1.16, 1.42] | $< 0.001$ |

# D  Claim of Artifacts

This study exclusively utilizes open-access resources, including publicly available datasets, interfaces (APIs), and established software libraries. Throughout the research process, we did not create or train any new machine learning models. Consequently, there are no proprietary or unpublished artifacts generated in this work that require submission. However, to facilitate future research and ensure reproducibility, our curated and cleaned datasets will be made available to interested parties upon reasonable request following the acceptance of this paper.

# E  Annotation and Classification Details

## E.1  LLM-Assisted Annotation

**Predefined Taxonomy.** Prior to LLM-assisted annotation, we established a predefined taxonomy through a structured human review process. Two annotators with expertise in ML security manually reviewed 300 papers in total. To balance coverage and efficiency, we adopted a tiered sampling strategy: for high-volume concept-topic combinations (those with more than 50 papers, including FL×DP, LLM×JB, LLM×DP, and FL×MIA), we randomly sampled 15% of papers for manual verification; for low-volume combinations (those with 20 or fewer papers), we reviewed all papers. Through iterative discussion and consensus building, the annotators refined the category boundaries and definitions, resulting in a taxonomy of 30 technique families.

**Annotation Methodology.** We use Claude Haiku 4.5 (via Google Vertex AI) to annotate all 1,591 paper abstracts. Each abstract is processed along with its ML concept and security topic labels. The LLM extracts two types of information: (1) the primary *technique family* from a predefined taxonomy of 30 categories, and (2) *threat model dimensions*, including access level (white-box / black-box / gray-box), attack phase (training-time / inference-time), data modality, and model access type (for LLM papers: open model / closed API). We additionally instruct the LLM to fill in "Others" if they believe it does not belong to any of our predefined technique families.

**Taxonomy Design.** After the LLM-assisted annotation, 34 entities are labeled as "Others." We then conduct a second-round human review, and the 34 entities were classified into two (empirical analysis and toolkit), yielding the final taxonomy of 32 families (Table 12). Attack families include: backdoor injection,

Table 12: Complete technique family taxonomy with definitions and paper counts. Families are grouped by type (attack, defense, analysis) and sorted by count within each group.

| Technique Family | Definition | Count |
|---|---|---|
| *Attack Techniques* | | |
| backdoor_injection | Plants a hidden trigger in the model during training, causing targeted misclassification when the trigger is present at inference time. | 216 |
| data_poisoning_general | Corrupts training data to degrade model performance or alter predictions, without using a specific trigger pattern (non-backdoor). | 150 |
| jailbreak_prompt_level | Uses carefully crafted natural language prompts to bypass LLM safety filters and elicit prohibited content. | 107 |
| membership_inference | Infers whether a specific data sample was used in the model's training set, posing a privacy threat. | 93 |
| adversarial_perturbation | Crafts small, often imperceptible perturbations to inputs that cause the model to produce incorrect predictions at inference time. | 63 |
| prompt_injection | Manipulates LLM input prompts to override system instructions and hijack model behavior for unintended purposes. | 33 |
| model_inversion | Reconstructs or approximates training data (e.g., images, text) from model outputs or parameters. | 20 |
| property_inference | Infers global properties of the training data distribution (e.g., class ratios, sensitive attributes) from model behavior. | 19 |
| transfer_attack | Generates adversarial examples on a surrogate model and transfers them to attack a different target model. | 18 |
| jailbreak_token_level | Uses token-level optimization (e.g., GCG, AutoDAN) to generate adversarial suffixes that bypass LLM safety mechanisms. | 18 |
| model_extraction | Steals model parameters, architecture, or functionality through query interactions with the target model. | 17 |
| other_attack | Attack techniques not covered by the above categories. | 8 |
| jailbreak_multi_turn | Uses multi-turn conversation strategies to gradually steer LLMs into producing prohibited content. | 5 |
| gradient_based_attack | Uses gradient information (e.g., from a white-box model) as the primary mechanism for constructing the attack. | 3 |
| *Defense Techniques* | | |
| anomaly_detection | Detects poisoned data samples, malicious clients (in FL), or other anomalous inputs to filter out threats. | 243 |
| robust_aggregation | Byzantine-robust aggregation methods for federated learning that tolerate malicious client updates. | 208 |
| differential_privacy | Applies differential privacy mechanisms (e.g., noise addition, gradient clipping) to protect data privacy. | 87 |
| adversarial_training | Augments training with adversarial examples to improve model robustness against adversarial perturbations. | 81 |
| input_preprocessing | Detects, transforms, or sanitizes potentially adversarial inputs before they reach the model at inference time. | 64 |
| federated_secure_aggregation | Uses secure computation techniques (MPC, homomorphic encryption, secret sharing) for privacy-preserving aggregation in FL. | 56 |
| safety_filter | Implements input/output filters for LLM safety, including toxicity detection, content moderation, and guardrails. | 37 |
| model_pruning_distillation | Uses model pruning or knowledge distillation as a defense mechanism to remove backdoors or reduce attack surfaces. | 23 |
| unlearning | Removes the influence of specific data points or concepts from a trained model without full retraining. | 19 |
| other_defense | Defense techniques not covered by the above categories. | 19 |
| watermarking | Embeds watermarks into model parameters or outputs for ownership verification and intellectual property protection. | 18 |
| certified_defense | Provides provable, mathematically guaranteed robustness bounds against specific classes of perturbations. | 14 |
| alignment_rlhf | Uses reinforcement learning from human feedback (RLHF) or similar alignment techniques to improve model safety. | 6 |
| *Analysis & Tooling* | | |
| empirical_analysis | Analyzes, compares, or evaluates existing attack/defense methods through experiments, without proposing a substantially new method. | 24 |
| toolkit | Provides a toolkit, framework, or platform for conducting attacks, defenses, or red-teaming evaluations. | 10 |

general data poisoning, adversarial perturbation, model extraction, membership inference, prompt injection, jailbreak (prompt-level / token-level / multi-turn), transfer attack, model inversion, property inference, and gradient-based attack. Defense families include: robust aggregation, anomaly detection, differential

privacy, adversarial training, certified defense, input preprocessing, model pruning/distillation, unlearning, alignment/RLHF, safety filter, watermarking, and federated secure aggregation.

**Quality Assurance.** On the 300 human-annotated papers, the LLM-assigned technique family matched the human judgment in over 93% of cases. The most common disagreement involved closely related families (e.g., `backdoor_injection` vs. `data_poisoning_general`), where the LLM sometimes chose the broader category. Note that the taxonomy table (Table 12) reports entry-level counts rather than unique paper counts; since some papers appear in multiple concept-topic combinations, entry counts may be slightly higher.

**Out-of-Sample Annotation.** The 93% figure above is the agreement between the LLM and the human labels on the 300 papers used to design the taxonomy. We note that Claude Haiku 4.5 was used zero-shot and was not fine-tuned on these papers, so this is not a case of evaluating a model on its own training data; the in-sample bias is therefore minor.

To further assess out-of-sample reliability, we drew a fresh random sample of 100 papers from the full corpus and re-annotated each one with three human annotators. The three human annotations agree at Fleiss' $\kappa = 0.905$, and the original haiku annotation matches the results of majority vote on 88% of papers, close to the 93% in-sample figure. Agreement is near perfect for the high-frequency families that drive our counts, and the small drop concentrates in inherently ambiguous boundaries (for example, anomaly detection versus robust aggregation, and safety filter versus input preprocessing).

### E.2 Venue Classification Rules

For the venue distribution analysis in Section 5.3, we classify publication venues into four categories using substring matching on venue names (case-insensitive):

- **Security venues**: Venues containing keywords such as "security and privacy," "CCS," "USENIX Security," "NDSS," "ACSAC," "ESORICS," "information forensics and security" (TIFS), "dependable and secure computing" (TDSC), "SaTML," "TrustCom," "PETS," or "RAID."

- **ML/AI venues**: Venues containing keywords such as "NeurIPS," "ICML," "ICLR," "AAAI," "IJCAI," "CVPR," "ICCV," "ECCV," "ACL," "EMNLP," "NAACL," "KDD," "WWW," "JMLR," "TMLR," or "ICASSP."

- **Unpublished/Preprint**: Venues containing "arXiv," "bioRxiv," "OpenReview," or "SSRN."

- **Other**: All remaining venues, including domain-specific journals and workshops not covered above.

Each paper is classified based on its primary venue field from Semantic Scholar. Papers without a venue field are classified as unpublished.

## F  Supplementary Technique Evolution Figures

This section presents the technique family evolution charts for all 17 concept×topic combinations with at least 20 papers. The four combinations discussed in the main text (FL×DP, LLM×JB, LLM×DP, LLM×PI) are shown in Figure 11. The remaining 13 combinations are shown below.

## G  Supplementary Tables and Figures

In this section, we present other supplementary figures and tables to provide additional support and clarity for our results.

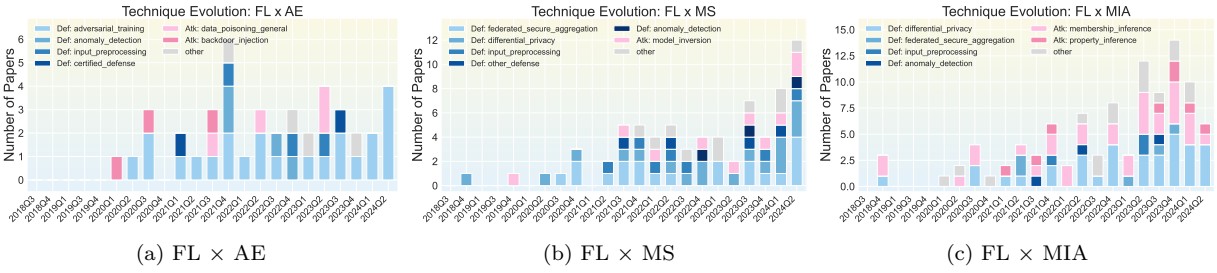

Figure 18: Technique evolution for FL (remaining combinations).

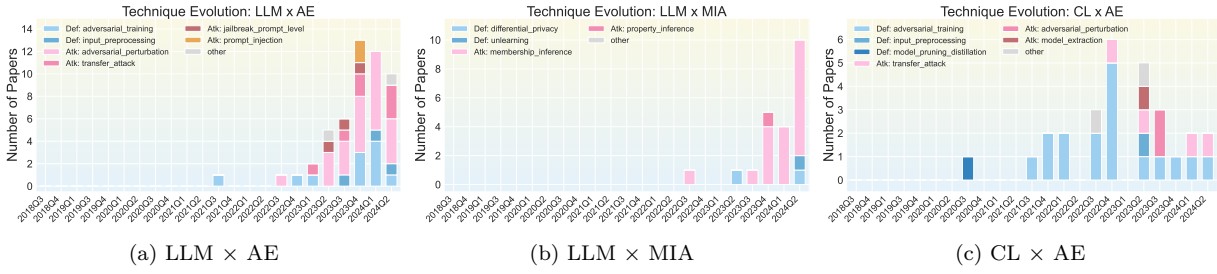

Figure 19: Technique evolution for LLM (remaining) and CL.

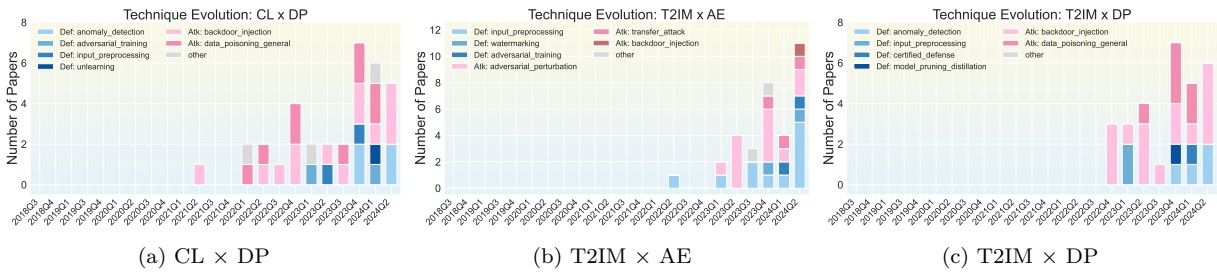

Figure 20: Technique evolution for CL (remaining) and T2IM.

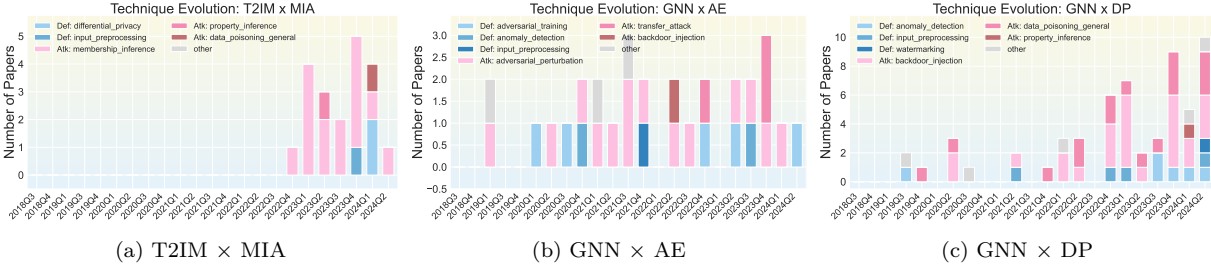

Figure 21: Technique evolution for T2IM (remaining) and GNN.

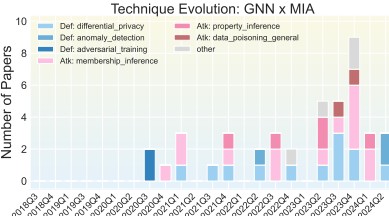

Figure 22: Technique evolution for GNN × MIA.

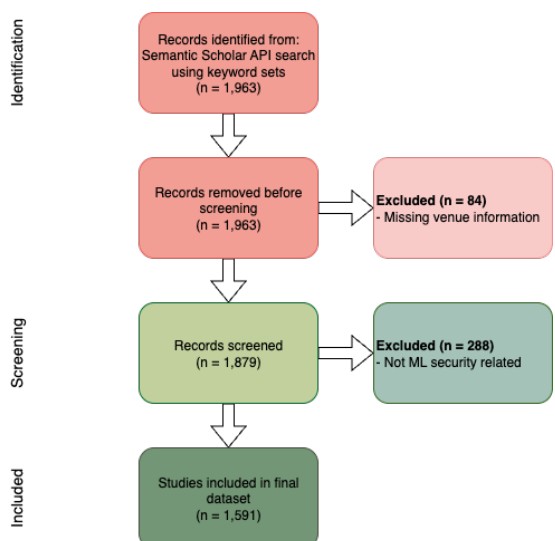

Figure 23: PRISMA flow diagram of the paper collection and screening process.

Table 13: An overview of the cleaned paper dataset's attributes.

| Attribute | Description | Data Source |
|---|---|---|
| Title | The title of a paper | Semantic Scholar |
| Abstract | The abstract of a paper | Semantic Scholar |
| Venue | The name of the paper's publication venue. | Semantic Scholar & DBLP |
| Available Date | Date of first public accessibility of a paper | Semantic Scholar |
| Author | The authors of a paper | Semantic Scholar |
| Citation Count | Count of times a paper is cited by other papers | Semantic Scholar |
| Institute | The institute or research unit to which the authors of the paper are affiliated. We focus only on specific universities or research institutes and do not record information about specific faculties or departments. | Human annotation |
| Region | The region where the institute is located. | Derived from "Institute" with (United Nations, 2024). |
| ML Concept | The ML concept targeted by a paper. Its value is in [*FL*, *CL*, *LLM*, *T2IM*, *GNN*]. | Human annotation |
| Security Topic | The security topic addressed in a paper. Its value is in [*AE*, *DB*, *MS*, *MIA*, *JB*, *PI*]. | Human annotation |
| Type | The main type of a paper. Its value is in [*attack*, *defense*, *both*]. | Human annotation |

Table 14: Summary statistics of citation density of security papers against five ML concepts.

| ML concept | Count | Mean | STD | Min | 25% | 50% | 75% | Max |
|---|---|---|---|---|---|---|---|---|
| FL | 858 | 0.019 | 0.049 | 0.000 | 0.000 | 0.006 | 0.016 | 0.585 |
| CL | 73 | 0.016 | 0.023 | 0.000 | 0.002 | 0.008 | 0.017 | 0.112 |
| LLM | 398 | 0.079 | 0.157 | 0.000 | 0.011 | 0.033 | 0.082 | 1.87 |
| T2IM | 92.0 | 0.028 | 0.043 | 0.000 | 0.004 | 0.014 | 0.039 | 0.344 |
| GNN | 148.0 | 0.016 | 0.029 | 0.000 | 0.000 | 0.006 | 0.016 | 0.191 |

Table 15: Fleiss' $\kappa$ inter-annotator agreement for the two human-annotated attributes. The "MLConcept" and "SecurityTopic" labels are derived from the query keyword pairs, and the threat-model dimensions are produced by LLM-assisted annotation (see Appendix E).

| Attribute | Metric Type | Fleiss' $\kappa$ |
|---|---|---|
| IfKeep | Binary | 0.754 |
| AttackOrDefense | Categorical | 0.730 |

Table 16: Summary statistics of citation density of security papers on six security topics.

| Security Topic | Count | Mean | STD | Min | 25% | 50% | 75% | Max |
|---|---|---|---|---|---|---|---|---|
| AE | 189 | 0.026 | 0.063 | 0.000 | 0.001 | 0.006 | 0.022 | 0.522 |
| DP | 842 | 0.021 | 0.044 | 0.000 | 0.001 | 0.008 | 0.020 | 0.490 |
| MS | 109 | 0.018 | 0.036 | 0.000 | 0.000 | 0.006 | 0.018 | 0.271 |
| MIA | 190 | 0.023 | 0.052 | 0.000 | 0.002 | 0.007 | 0.022 | 0.585 |
| JB | 201 | 0.112 | 0.203 | 0.000 | 0.023 | 0.051 | 0.112 | 1.872 |
| PI | 38 | 0.070 | 0.107 | 0.000 | 0.009 | 0.026 | 0.089 | 0.433 |

Table 17: Summary statistics of citation density of security papers with different author counts.

| Author | Count | Mean | STD | Min | 25% | 50% | 75% | Max |
|---|---|---|---|---|---|---|---|---|
| 1 | 19 | 0.008 | 0.014 | 0.000 | 0.000 | 0.002 | 0.010 | 0.044 |
| 2 | 153 | 0.016 | 0.038 | 0.000 | 0.000 | 0.004 | 0.013 | 0.250 |
| 3 | 251 | 0.028 | 0.094 | 0.000 | 0.001 | 0.007 | 0.021 | 1.192 |
| 4 | 327 | 0.038 | 0.125 | 0.000 | 0.003 | 0.009 | 0.025 | 1.872 |
| 5 | 267 | 0.026 | 0.051 | 0.000 | 0.002 | 0.009 | 0.026 | 0.524 |
| 6 | 222 | 0.040 | 0.092 | 0.000 | 0.003 | 0.013 | 0.039 | 0.916 |
| 7 | 132 | 0.037 | 0.090 | 0.000 | 0.005 | 0.014 | 0.035 | 0.896 |
| 8 | 55 | 0.052 | 0.072 | 0.000 | 0.007 | 0.029 | 0.071 | 0.379 |
| 9+ | 55 | 0.083 | 0.131 | 0.000 | 0.007 | 0.034 | 0.093 | 0.659 |

Table 18: Summary statistics of citation density of security papers with top 6 affiliated regions.

| Region | Count | Mean | STD | Min | 25% | 50% | 75% | Max |
|---|---|---|---|---|---|---|---|---|
| AUS | 113 | 0.036 | 0.081 | 0.000 | 0.004 | 0.011 | 0.035 | 0.659 |
| CA | 67 | 0.021 | 0.036 | 0.000 | 0.002 | 0.007 | 0.020 | 0.193 |
| CN | 697 | 0.028 | 0.062 | 0.000 | 0.001 | 0.009 | 0.029 | 0.659 |
| FR | 36 | 0.016 | 0.037 | 0.000 | 0.000 | 0.005 | 0.015 | 0.206 |
| DE | 91 | 0.062 | 0.204 | 0.000 | 0.006 | 0.017 | 0.049 | 1.872 |
| IN | 32 | 0.025 | 0.063 | 0.000 | 0.000 | 0.003 | 0.013 | 0.271 |
| JPN | 44 | 0.034 | 0.066 | 0.000 | 0.000 | 0.010 | 0.031 | 0.379 |
| SGP | 78 | 0.062 | 0.114 | 0.000 | 0.006 | 0.023 | 0.065 | 0.659 |
| KOR | 42 | 0.017 | 0.050 | 0.000 | 0.002 | 0.005 | 0.011 | 0.317 |
| UK | 100 | 0.057 | 0.195 | 0.000 | 0.005 | 0.014 | 0.038 | 1.872 |
| USA | 531 | 0.059 | 0.142 | 0.000 | 0.005 | 0.016 | 0.050 | 1.872 |

Table 19: Summary statistics of citation density of security papers with different collaboration patterns.

| Pattern | Count | Mean | STD | Min | 25% | 50% | 75% | Max |
|---|---|---|---|---|---|---|---|---|
| Domestic | 486 | 0.034 | 0.078 | 0.000 | 0.003 | 0.009 | 0.028 | 0.896 |
| Independent | 505 | 0.026 | 0.081 | 0.000 | 0.000 | 0.007 | 0.022 | 1.192 |
| International | 484 | 0.043 | 0.115 | 0.000 | 0.004 | 0.012 | 0.039 | 1.872 |

Table 20: Summary statistics of citation density of security papers with different publication status.

| Status | Count | Mean | STD | Min | 25% | 50% | 75% | Max |
|---|---|---|---|---|---|---|---|---|
| Published | 1013 | 0.031 | 0.077 | 0.000 | 0.002 | 0.009 | 0.026 | 1.192 |
| Unpublished | 469 | 0.041 | 0.118 | 0.000 | 0.003 | 0.011 | 0.035 | 1.872 |

Table 21: Defense lag (in quarters) using target-only entries. Positive values indicate defense trailing attack; negative values indicate defense preceding attack (proactive defense or overlapping topic boundaries). "—" indicates insufficient data.

| Topic \ Concept | FL | CL | LLM | T2IM | GNN |
|---|---|---|---|---|---|
| AE | 1 | −4 | −4 | −1 | 0 |
| DP | −1 | 7 | 5 | 1 | −1 |
| MS | −13 | — | −2 | — | 8 |
| MIA | 0 | 0 | 3 | 2 | −1 |
| JB | — | — | 1 | 4 | — |
| PI | — | — | 1 | — | — |

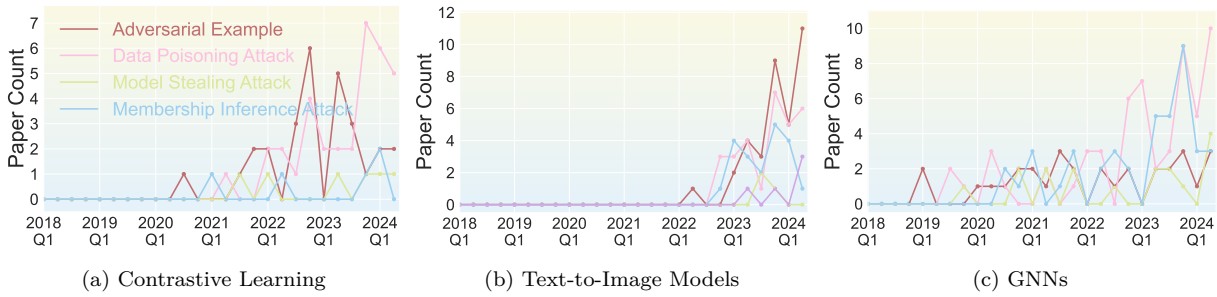

Figure 24: Evolution of the number of security papers for CL, T2IMs, and GNNs. The y-axis represents the number of papers, while the x-axis indicates time, with each data point showing the total number of papers published within a three-month period (a quarter).

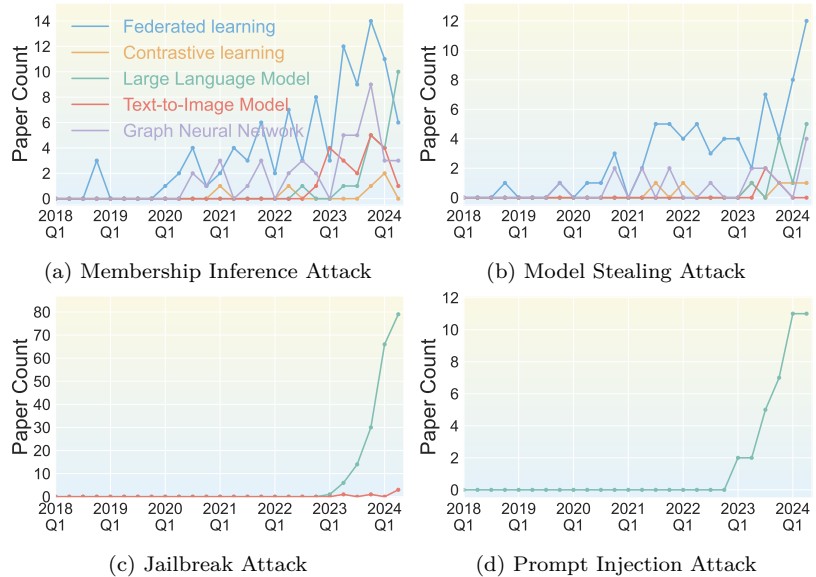

Figure 25: Evolution of the number of security papers for model stealing, MIA, jailbreak, and prompt injection. The y-axis represents the number of papers, while the x-axis indicates time, with each data point showing the total number of papers published within a three-month period (a quarter).

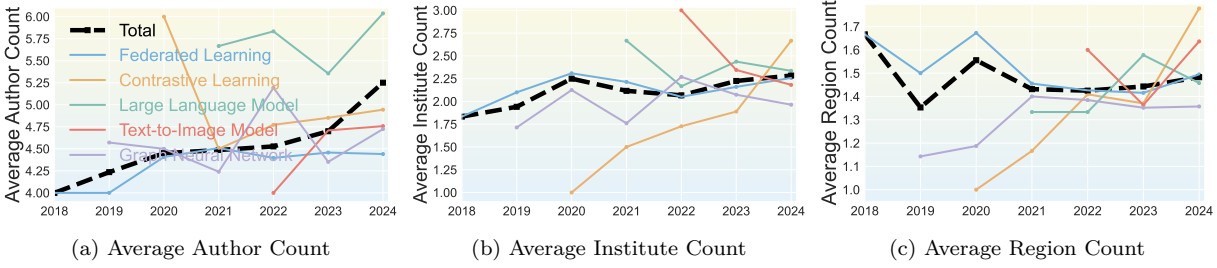

Figure 26: The Evolution of Average Author Count, Institute Count, and Region Count for Security Papers of 5 ML Concepts

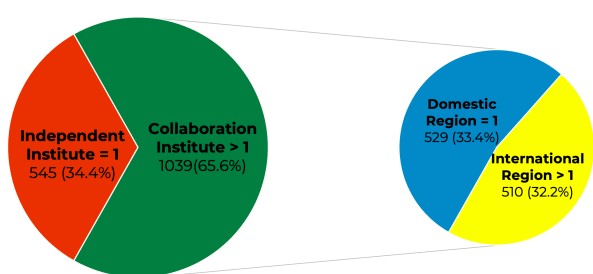

Figure 27: Proportion of three collaboration patterns.

