# OpenReview forum: "A Meta-Analysis of Machine Learning Security Research: Attack-Defense Dynamics, Technical Evolution, and Cross-Concept Patterns"
_TMLR — Decision pending for TMLR_

### Review · Reviewer_EGSK · 2026-05-19

**Summary Of Contributions:**

This paper presents a data-driven meta-analysis of 1,591 ML security papers published between January 2018 and June 2024, systematically mapping the intersection of five ML paradigms (FL, CL, LLMs, T2IMs, and GNNs) with six major security topics (Adversarial Examples, Data Poisoning, Model Stealing, Membership Inference, Jailbreaks, and Prompt Injection). It further highlights key trends, including the attack–defense imbalance, shifts in threat models, and the technical evolution of the field.

**Audience:**

Yes

**Audience Explanation:**

The results are quite reasonable and effectively capture the domain shift over time. The paper is well-structured, the methodology for data collection (via Semantic Scholar/DBLP and PRISMA guidelines) is sound, and the visualizations effectively communicate the core narrative.

**Claims And Evidence:**

Yes

**Claims Explanation:**

* In Section 5.4, the authors use "citation density" (citations per day) to compare the academic influence of various factors, concluding that unpublished papers and LLM papers have higher density. However, this metric is inherently skewed toward rapidly growing, recent fields currently experiencing a "citation burst" (like LLMs post-ChatGPT). Older papers outside the current mainstream (e.g., 2018 FL papers) will naturally have a flattened daily average over 5-6 years. Because LLM papers constitute the bulk of recent preprints and exist in this citation burst phase, the claim that "unpublished papers tend to have slightly higher citation density" is heavily confounded by temporal bias and topic popularity.
* In Section 3.4, the authors quantify a "defense lag" and highlight the severe imbalance between attack and defense papers. While the data is accurate, the discussion lacks the fundamental context of why this lag is scientifically reasonable and expected. There is an asymmetric burden of proof in ML security research: proving an attack is valid generally only requires demonstrating its success on specific models or within a specific scenario. Conversely, introducing a defense requires rigorous proof that the method can withstand numerous, diverse attacks and generalize well across different architectures.

**Requested Changes:**

* The authors should explicitly discuss the limitations and temporal bias of citation density across drastically different time horizons. To strengthen the claims in Section 5.4, the authors should either conduct a multivariate analysis (controlling for publication year and ML Concept) to isolate the true effect of publication status, or clearly caveat that the high citation density is likely driven by the "LLM hype cycle" rather than preprint status intrinsically.
* Expand the discussion of the "defense lag" to explicitly acknowledge this structural asymmetry. Contextualizing the metric by discussing the higher barrier to entry and generalizability requirements for defense papers will provide a richer, more accurate picture of why defense research inherently takes longer to develop and publish.

---

### Review · Reviewer_PmYK · 2026-05-24

**Summary Of Contributions:**

-This paper reports on an analysis of almost 1,600 machine learning security research papers. So it is simultaneously a paper about methods in the field, but also about the field itself, which I think is interesting. The authors scrape different types of machine learning papers on different types of security topics and then use a mix of human annotation and language model annotation to create a rich data set of information which they then analyze. This leads to some broad field-wide information about what kinds of papers are published, what the attack versus defense offense is, and how the field has been evolving in the last few years.

**Additional Comments:**

None

**Audience:**

Yes

**Audience Explanation:**

Yeah, I think this is absolutely the case. The paper is doing a pretty unique and valuable thing in distilling field-wide lessons from over 1500 research papers.

**Claims And Evidence:**

No

**Claims Explanation:**

I'm very very suspicious and unsatisfied with the selection of six security topics. I think there are a handful of problems. One of which is that adversarial examples have a ton of overlap with other topics like jail breaks and prompt injection. I can't find a definition of these things that disentangles them somewhere in the paper. Some sort of justification for why these six things are disjoint and some sort of definition for what they are seems necessary, but if those things were provided, I think it would be troublesome because these are six terms that people play really fast and loose with in the research field. Overall, I think that, for example, Figure 3A tells me absolutely nothing useful about the research field. It just tells me some really muddy picture that is influenced by the particular operational semantics that this paper ascribes to these terms.

I'm kind of perplexed by looking at the counts of papers that were surfaced under the different ML concepts. For example, I know that there are far more than 76 contrastive learning papers out there from the last few years, and far more than 95 text to image model papers related to security. Based on this, I am somewhat inclined to infer that the paper collection methodology wasn't very great. But somewhat more confidently, I definitely think it's not a good idea to try to present something like Figure 2. Because Figure 2 seems like it's saying something about the field overall when it's not doing a good job of that. It's just saying something about the data set collected.

I think there's a mismatch between what the paper claims and what the defense ratio actually measures. The defense ratio just measures paper counts, but is not able to measure anything about how strong attacks are versus defenses. It is entirely possible to, for example, write ten papers introducing great defenses that a hundred bad papers on weak attacks cannot break. I have related concerns about the defense lag measure. It's also really easy to see how this can be decoupled from the practical state of the offense-defense balance in the field in a lot of different ways. Overall, I'm starting to think that this paper has done some significant overclaiming without engaging better with the validity problems of some of these measures.

I have some more skepticism about what's being claimed in section 4.4. I think there's some more overclaiming here and I think there is some implicit and not so great assumption that the count of papers on a certain topic, tell us reliable information about the vulnerabilities of AI systems with respect to that topic. Counting the number of papers on something just seems like a really poor proxy in a lot of ways.

**Requested Changes:**

The machine learning concepts and security topics in Tables 1 and 2 are not justified. I didn't see a part of the paper explaining why those topics and only those topics were an exhaustive list or the right list to choose. So I think the authors should add a good discussion of why and how they selected those things.

Overall, I think that to accept a paper like this, I would roughly recommend getting rid of sections 3.2, 3.3, and 4.4. And I would work through the paper to confront the validity problems with some of the measures taken over the dataset and also the representativeness problems of the dataset and reduce overclaiming accordingly.

---

### Review · Reviewer_zkfj · 2026-05-26

**Summary Of Contributions:**

The paper presents an empirical meta-analysis of 1,591 ML-security papers (Jan 2018 – Jun 2024) collected from Semantic Scholar + DBLP, covering five ML "concepts" (Federated Learning, Contrastive Learning, LLMs, Text-to-Image Models, Graph Neural Networks) and six security topics (adversarial examples, data poisoning, model stealing, membership inference, jailbreak, prompt injection). Three research questions are addressed:

- **RQ1 (landscape):** quarter-by-quarter publication trends; an "attack-defense imbalance" measured by a *defense ratio* (defense papers / total) per concept×topic cell; a "defense lag" in quarters between the first attack and the first defense paper.
- **RQ2 (techniques):** LLM-assisted (Claude Haiku 4.5) abstract annotation into a 32-family taxonomy; temporal evolution of those families; identification of 17 cross-concept shared families and 6 spanning all five concepts; threat-model decomposition along access (white-/gray-/black-box) and phase (training/inference).
- **RQ3 (influence):** Kruskal-Wallis/Mann-Whitney + Dunn's post-hoc tests on citation density against six factors (concept, topic, authors, region, collaboration, publication status).

**Key strengths.**
- Wide scope (5×6 grid) and explicit PRISMA-style data-collection pipeline.
- Sizeable human-annotation effort (5 experts, 3 labelers/paper, 300+ person-hours, Fleiss' κ = 0.742).
- The defense-ratio + defense-lag pairing is a useful and clear framing of attack/defense asymmetry.
- The five-cell access × phase view (Table 4) and the cluster A/B contrast (FL-dominated white-box/training vs. LLM-dominated black-box/inference) is the most original observation in the paper.
- Limitations section is candid about keyword-coverage and annotator subjectivity.

**Key weaknesses.**
- Several core constructs (defense ratio, "first appearance," "shared technique family") are sensitive to the search window, keyword sets, and abstract-only annotation in ways the paper does not fully quantify.
- The RQ3 statistical pipeline is univariate, ignores known confounders (especially time × topic), reports no effect sizes, and applies multiplicity correction only inside Dunn's test (not across six family-wise tests).
- The LLM annotation accuracy (93 %) is reported on the same 300-paper set used to build the taxonomy, so it does not generalize-test the labels.
- Keyword sets are narrow (LLM excludes `RLHF`, `alignment`, `red-teaming`, `instruction tuning`, etc.), and selection bias is plausibly entangled with the headline imbalance findings.
- No Broader Impact Statement despite the dual-use nature of the surveyed area.

Overall the paper is a useful descriptive resource for ML-security researchers, but several quantitative claims need either more defensive analysis or softer wording.

**Audience:**

Yes

**Audience Explanation:**

A nontrivial subset of TMLR readers are ML security researchers.

**Claims And Evidence:**

No

**Claims Explanation:**

Partial – claims are mostly supported as descriptive observations on the assembled dataset, but several causal/comparative claims and the RQ3 statistical conclusions are not adequately defended.

**Requested Changes:**

**Critical (required for acceptance):**

C1. **Defense-ratio robustness.** Add a sensitivity table showing how the per-cell defense ratios in Fig. 7 change under (i) excluding "Both" papers, (ii) counting "Both" only as defense, (iii) counting "Both" only as attack. Re-state the headline rankings only for those cells that remain stable under all three rules. The current rule (count toward both) is defensible but its consequences for cross-cell ranking are not shown.

C2. **Independent LLM-annotation validation.** Hold out an additional independent sample (e.g., 100 papers stratified by concept×topic) that was *not* used in taxonomy construction, re-annotate manually, and report Claude Haiku 4.5 accuracy + per-family confusion matrix on this held-out set. The current 93 % figure is on the taxonomy-design set and cannot be cited as out-of-sample accuracy.

C3. **RQ3 multivariable analysis.** Re-do the citation-influence analysis with a multivariable model (Poisson or negative-binomial GLM on citation counts with `log(days_since_publication)` as offset; predictors = concept, topic, authors, region, collaboration, publication status). Report coefficient estimates with 95 % CIs and effect sizes, not only p-values. Either retain or attenuate the §5.5 narrative based on the multivariable result. At minimum, the §5.5 "topic selection matters more than other factors" and "preprint > published" claims must be qualified or removed if the multivariable model attributes them to topic-recency confounding.

C4. **Keyword-set recall check.** Estimate the recall of the keyword sets by (a) cross-checking against the program of one major security venue per year (e.g., USENIX Security, NDSS, S&P, CCS) and computing the fraction of in-scope papers that were retrieved, or (b) using a small expanded keyword set on a single year and reporting how counts move. This is necessary to defend cross-cell counts that drive the imbalance narrative — especially comparisons involving CL (single keyword) and JB (single keyword).

C5. **First-appearance vs. first-in-corpus.** Throughout §4.3 and §6, replace "first appears" with "first appears in our corpus (Jan 2018 onward)" or equivalent. The current language repeatedly suggests a technical lineage claim that the data cannot support.

C6. **Numerical reconciliation.** Add a small "counting conventions" box reconciling: (i) 1,591 unique papers vs. 1,682 concept×topic entries; (ii) Table 4 cell sums vs. 1,591; (iii) Table 5 (unique-paper counts) vs. Table 10 (entry counts); (iv) regional counts that exceed 1,591 because of multi-region attribution. Each table caption should explicitly state which counting convention applies.

C7. **Per-cell annotator agreement.** Report Fleiss' κ broken down by label (`IfKeep`, `MLConcept`, `SecurityTopic`, `AttackAccess`, `AttackPhase`) rather than only the aggregate κ = 0.742. The defense ratio is sensitive to per-paper attack/defense labeling, so its agreement specifically matters.

C8. **Reference / citation hygiene.** Several in-text citations have mismatched author names and cite-keys, e.g., "Biggio and Roli Guan et al. (2018)" (§8), "Kairouz et al. Kuntla et al. (2021)" (§8), "Chakraborty et al. Chowdhury et al. (2024)" (§8). URL-only references (Sem, DBL, gdp, pri, ube, goo) need proper bibliographic entries or a clearly labeled "online resources" section.

C9. **Address the data-recency gap.** The data cutoff is June 30, 2024; at the time of this review (May 2026) that is nearly two years stale in a field whose own §3 figures show roughly a 2× increase per year. The authors should pursue **one** of the following before acceptance:

   (a) *Extend the dataset.* Re-run the Semantic Scholar + DBLP collection through at least Dec 31, 2025 and re-compute Table 3, Figure 7 (defense ratio), Table 4 (threat-model clusters), and the §5.4 citation-density tests. The pipeline is automated, so the marginal cost is bounded; the change in headline numbers is informative on its own. **Or**

   (b) *Re-position as an explicit retrospective snapshot.* Change the framing in §1, §6, and §9 from "concrete directions for future research" to "a Jan 2018 – Jun 2024 baseline for future longitudinal comparison." Add a §6.x "Developments Since the Data Cutoff" subsection noting at least the post-cutoff shifts that would materially update the paper's specific numerical conclusions, including:

   - **New technique families absent from the taxonomy:** agentic / tool-use attacks; RAG and memory poisoning; fine-tuning-API attacks; sleeper-agent / alignment-faking research; many-shot and best-of-N jailbreaks; reasoning-model-specific attacks (chain-of-thought injection, deliberative-alignment bypass); etc.
   - **Defense-ratio shifts:** the "LLM × JB defense ratio = 0.30" headline almost certainly understates 2025–2026 defense activity (circuit breakers, deliberative alignment, R2D2, latent-space and refusal-training defenses), and should be flagged as a 2024 H1 snapshot rather than current state.
   - **FL vs. LLM volume inversion:** §3 already shows LLM overtaking FL in Q1 2024; the FL-centric weight throughout the paper (gray-box analysis, mature-defense narrative) over-represents the current landscape and should be qualified.
   - **Citation-density volatility:** §5 conclusions are particularly sensitive to two additional years of accumulation; either re-snapshot or explicitly mark the 2024 H1 numbers as no-longer-current.

   Option (a) is preferred; option (b) is acceptable if the data-collection pipeline cannot be re-run on the original infrastructure. What is *not* acceptable is leaving the paper as-is with the current forward-looking framing, because readers will treat the 2024 H1 numbers as a present-day map of where to invest defense effort.

   The structural observations of the paper — the white-box-training vs. black-box-inference cluster contrast, the cross-concept paradigm-independent threats, and the FL/GNN-as-early-adopter pattern — are largely time-invariant and survive the recency gap; this request is about framing and headline numbers, not about the analytical core.

**Strengthening (would improve the work but not strictly required):**

S1. **Multi-label technique annotation.** Many papers propose both an attack and a defense, or use multiple techniques. Replace the "primary technique family" single-label scheme with multi-label and re-report Tables 5/6/10 on a counted-per-paper basis with explicit assumptions.

S2. **Full-text vs. abstract-only annotation comparison.** On the held-out validation set in C2, also annotate from the full text and report the disagreement rate vs. abstract-only annotation. This bounds the systematic bias from using abstracts.

S3. **Publication-year-controlled defense ratios.** Plot defense ratio per cell over time (paralleling Fig. 10) — this would directly test whether the LLM gap is widening or merely a leading-edge effect that closes with maturity.

S4. **Citation-density discussion.** Move the caveat about citation density being highly sensitive to publication-age dynamics into §5.4.1, and explicitly acknowledge that "preprint > published" is largely driven by LLM × time confounding.

S5. **Open dataset release.** TMLR encourages reproducibility. Anonymously deposit the curated 1,591-paper metadata + annotation file (paper IDs, labels, citation counts at snapshot date) at submission rather than "available upon reasonable request following acceptance." This would let reviewers and future readers re-run the central analyses.

S6. **Use of LLM for annotating LLM-security papers.** Add a short paragraph addressing the methodological reflexivity: an LLM-based annotator (Claude Haiku 4.5) is itself an instance of the systems being classified, and may have stronger priors on LLM-related techniques (e.g., `safety_filter`, `alignment_rlhf`, `jailbreak_*`). Show whether per-family accuracy differs between LLM-related and non-LLM-related papers in the C2 validation set.

S7. **Clarify "ML concept" assignment.** §2.1 states that for a paper using CL to attack T2IM, the concept is "T2IM" not "CL." Spell out the assignment algorithm (annotator rule + tie-breaking) and report how often annotators disagreed on this specifically. This affects Table 3 distributions directly.

S8. **Threat-model categorisation transparency.** Provide the actual Claude Haiku prompt used to classify access level and phase, and a sample of borderline cases (e.g., a paper that fine-tunes a closed LLM via API — white-box or black-box?). Currently App. E.1 describes the protocol but does not show the prompt.

S9. **Defense-lag interpretation.** §3.4 acknowledges that negative defense lags often reflect *labeling drift* rather than proactive defense (e.g., DP for FL existed before MS attacks against FL were studied). This is the right interpretation; the discussion in §3.5 ("proactive defense design is more effective than reactive response") overstates it. Tighten the §3.5 language so it is consistent with the §3.4 interpretation.

S10. **Typographical.** "x1 = ML concepts" through "x6 = publication status" — italicise. "α = 0.05" appears with an unrendered Greek letter in several places. "DBLP DBL" / "Semantic Scholar Sem" with stray cite-keys read awkwardly.